# Measurement report: Nitrogen isotopes ($\delta^{15}$N) and first quantification of oxygen isotope anomalies ($\Delta^{17}$O, $\delta^{18}$O) in atmospheric nitrogen dioxide

**Sarah Albertin**[1,2], **Joël Savarino**[2], **Slimane Bekki**[1], **Albane Barbero**[2], and **Nicolas Caillon**[2]

[1] LATMOS/IPSL, Sorbonne Université, UVSQ, CNRS, 75005 Paris, France
[2] IGE, Univ. Grenoble Alpes, CNRS, IRD, Grenoble INP, 38000 Grenoble, France

*Correspondence to*: Sarah Albertin (sarah.albertin@latmos.ipsl.fr)

**Abstract.** The isotopic composition of nitrogen and oxygen in nitrogen dioxide ($NO_2$) potentially carries a wealth of information about the dynamics of the nitrogen oxides ($NO_x$ = nitric oxide (NO) + $NO_2$) chemistry in the atmosphere. While nitrogen isotopes of $NO_2$ are subtle indicators of $NO_x$ emissions and chemistry, oxygen isotopes are believed to reflect only the $O_3$/$NO_x$/VOC chemical regime in different atmospheric environments. In order to access this potential tracer of the tropospheric chemistry, we have developed an efficient active method to trap atmospheric $NO_2$ on denuder tubes and measured, for the first time, its multi-isotopic composition ($\delta^{15}$N, $\delta^{18}$O, and $\Delta^{17}$O). The $\Delta^{17}$O values of $NO_2$ trapped at our site in Grenoble, France, show a large diurnal cycle peaking in late morning at (39.2 ± 1.7) ‰ and decreasing at night until (20.5 ± 1.7) ‰. On top of this diurnal cycle, $\Delta^{17}$O also exhibits substantial daytime variability (from 29.7 to 39.2 ‰), certainly driven by changes in the $O_3$ to peroxyl radicals ($RO_2$) ratio. The nighttime decay of $\Delta^{17}$O($NO_2$) appears to be driven by $NO_2$ slow removal, mostly from conversion into $N_2O_5$, and its formation from the reaction between $O_3$ and freshly emitted NO. As expected from a nighttime $\Delta^{17}$O($NO_2$) expression, our $\Delta^{17}$O($NO_2$) measured towards the end of the night is quantitatively consistent with typical values of $\Delta^{17}$O($O_3$). Daytime N isotope fractionation is estimated using a general expression linking it to $\Delta^{17}$O($NO_2$). An expression is also derived for the nighttime N isotope fractionation. In contrast to $\Delta^{17}$O($NO_2$), $\delta^{15}$N($NO_2$) measurements exhibit little diurnal variability (−11.8 to −4.9 ‰) with negligible isotope fractionations between NO and $NO_2$, mainly due to high $NO_2$/$NO_x$ ratios, excepted during the morning rush hours. The main $NO_x$ emissions sources are estimated using a Bayesian isotope mixing model, indicating the predominance of traffic emissions in this area. These preliminary results are very promising for using the combination of $\Delta^{17}$O and $\delta^{15}$N of $NO_2$ as a probe of the $NO_x$ sources and fate and for interpreting nitrate isotopic composition records.

## 1 Introduction

Nitrogen oxides ($NO_x$ = $NO_2$ + NO) are at the heart of tropospheric chemistry, as they are involved in key reaction chains governing the production and destruction of compounds of fundamental interest for health, ecosystems and climate issues (Brown, 2006; Finlayson-Pitts and Pitts, 2000; Jacob, 1999). For example, $NO_2$ photolysis followed by reaction of NO with peroxy radicals ($RO_2$ = $HO_2$ + $RO_2$) is the only significant source of ozone ($O_3$) in the troposphere where it serves as a severe air pollutant and a greenhouse gas. Tropospheric $O_3$ also plays a major role in the production processes of radicals which are

responsible for the oxidation and removal of compounds emitted into the atmosphere (Crutzen, 1996). This "cleaning" ability is referred to as the atmospheric oxidative capacity (AOC; Prinn, 2003). Additionally, $NO_x$ species are at the core of the reactive nitrogen cycle as precursors of atmospheric nitrate (particulate $NO_3^-$ + gaseous $HNO_3$) which contributes to soil

acidification and eutrophication (Galloway et al., 2004), and aerosol radiative forcing (Liao and Seinfeld, 2005). In order to better understand the reactive nitrogen (which includes $NO_x$ and $HNO_3$) chemistry, the related AOC, and the contributions of precursors emissions to nitrate deposition, it is necessary to better constrain $NO_x$ emissions sources and individual oxidation processes.

Stable isotope analysis is a powerful tool for tracing emission sources, individual chemical mechanisms and budgets of

atmospheric trace gases (Kaye, 1987). Because physico-chemical and biological processes favour lighter or heavier isotopologues, the isotopic composition of a chemical species will often vary according to its formation pathway. This phenomenon of isotopic fractionation can thus be used to trace different processes involved in the formation of the chemical species being analyzed. Isotopic enrichment ($\delta$) of an element X is expressed in ‰ and defined as: $\delta^n X = (\,^n R_{spl} / \,^n R_{ref} - 1)$ with $^n R$ the elemental isotope abundance ratio of the heavy isotope over the light isotope (e.g. for oxygen isotopes $^{18}R(^{18}O/^{16}O)$

$\equiv \,^{18}R = x(^{18}O)/x(^{16}O)$ or $^{17}R(^{17}O/^{16}O) \equiv \,^{17}R = x(^{17}O)/x(^{16}O)$, with $x$ the isotopic abundance) in a sample ($^n R_{spl}$) and in a reference ($^n R_{ref}$). The Vienna Standard Mean Ocean Water (VSMOW; Li et al., 1988) and atmospheric nitrogen ($N_2$; Mariotti, 1984) are the international references for oxygen and nitrogen ratios, respectively. Most natural isotopic fractionations are mass dependent fractionations (MDF; Urey, 1947), as it is notably the case for terrestrial oxygenated species in which the triple oxygen composition follows $\delta^{17}O \approx 0.52 \times \delta^{18}O$ (Thiemens, 1999). Yet, laboratory experiments (Thiemens and Heidenreich,

1983) and atmospheric observations (Johnston and Thiemens, 1997; Krankowsky et al., 1995; Vicars and Savarino, 2014) have showed that the isotopic composition of ozone formed in the atmosphere does not follow this canonical MDF relationship and reflects mass independent fractionation (MIF) processes. The important deviation from the MDF oxygen relationship is called the oxygen-17 anomaly ($\Delta^{17}O$) and is defined here in its approximate linearized form as $\Delta^{17}O = \delta^{17}O - 0.52 \times \delta^{18}O$. Our choice of this linear definition is mainly motivated by its convenience for mass balance calculations and its validity for our

large $\Delta^{17}O$ values and variability. Overall, biases related to our choice of the linear definition are marginal in our conditions (Assonov and Brenninkmeijer, 2005). It follows that $\Delta^{17}O$ inherited from ozone can be considered as conserved during MDF processes.

The multi-isotopic composition of $NO_x$ is therefore a very valuable tracer of its emissions and chemistry in the atmosphere. However, so far, $\Delta^{17}O$ of atmospheric $NO_2$ ($\Delta^{17}O(NO_2)$) has been investigated only using laboratory (Michalski et al., 2014)

and modelling (Alexander et al., 2020, 2009; Lyons, 2001; Morin et al., 2011) approaches with theoretical frameworks, and these results need to be tested against atmospheric observations. Walters et al. (2018) have presented a method of sampling and analysing nitrogen and oxygen stable isotopes of $NO_2$ collected separately at daytime and nighttime in an urban area but they did not report on $\Delta^{17}O$. Dahal and Hastings (2016) have attempted to measure $\Delta^{17}O$ of $NO_2$ collected on passive samplers, but the isotopic signal was partly degraded during the sampling and the analytical procedure. Building on their work, we

present here an efficient method to collect atmospheric $NO_2$ for isotopic analysis and present the first measurements of triple oxygen isotopes and double nitrogen isotopes of atmospheric $NO_2$. Combined with mass-balance equations, oxygen isotopes are used to investigate the links between the variability of the oxygen isotope anomaly of $NO_2$ and its formation pathways. We also revisit the Morin et al. (2011) $NO_x$ isotopic theoretical framework and extend it to urban environments. After estimating the nitrogen isotopic fractionation between NO and $NO_2$, we infer from $\delta^{15}N$ of the $NO_2$ ($\delta^{15}N(NO_2)$) the major emission sources of $NO_x$ influencing our sampling site using an isotopic mixing model (Parnell et al., 2010).

## 2 Materials and Methods

### 2.1 Sampling method

$NO_2$ was sampled on an active (pumped) collection system using denuder tubes. This method is more efficient to collect $NO_2$ than passive methods (Røyset, 1998), allowing for shorter collection times with a breakthrough of the absorption capacity below 1 % (Buttini et al., 1987; Williams and Grosjean, 1990). The sampled air was pumped through a ChemComb$^{TM}$ 3500 speciation cartridge (Thermo Scientific$^{TM}$, USA). Initially used for the speciation of gases and aerosols, these advanced sampling platforms consist of a $PM_{2.5}$ impactor inlet connected to a stainless-steel cylinder that contains two glass honeycomb denuders connected in series for gas collection, and a Teflon stage filter pack for aerosols. To collect $NO_2$, glass tubes were coated with an alkaline guaiacol solution. In basic medium, guaiacol (IUPAC name: 2-Methoxyphenol) is known to react with $NO_2$ to form stable $NO_2^-$ ions (Nash, 1970) preserving the original $NO_2$ isotopic signal due to the basic nature of the medium (pH = 14 after 10 ml extraction). Because NO or peroxyacetyl nitrate (PAN) are not collected by guaiacol, this methodology avoids potential interferences from these compounds in later analyses (Buttini et al., 1987). Although nitrous acid (HONO) can bind as $NO_2^-$, it is unlikely to adversely impact the results as its concentration is much lower than $NO_2$ (by a factor of 10 to 20) even in very polluted cities (e.g., Harris et al., 1982; Michoud et al., 2014; Huang et al., 2017).

To evaluate the sampling system performance, a series of experiments were run with artificial gaseous $NO_2$. Using a commercial gas standard generator (KinTek FlexStream$^{TM}$) feed with zero-air, diluted $NO_2$ (Metronics Dynacal$^{TM}$) was sent through a ChemComb cartridge while $NO_x$ concentration was measured up- and down-stream of the cartridge. From 1 to 30 nmol mol$^{-1}$ of $NO_2$ (representative of rural to urban atmospheric conditions), concentrations coming out of the cartridge were never above the noise level of the $NO_x$ monitor (2.5 nmol mol$^{-1}$). To estimate the denuders trapping efficiency, we passed different concentrations of gaseous $NO_2$ through the collection apparatus and measured the amount of $NO_2^-$ collected on the two denuders both connected in series. The denuder efficiency $E$ was then calculated according to the following equation (Buttini et al., 1987):

$$E = (1 - \frac{b}{a}) \times 100 \text{ \%} \tag{1}$$

with $a$ and $b$ representing the amount of $NO_2^-$ collected on the first and the second denuder, respectively. From 0.3 to 1.3 µmol of generated $NO_2$ (see Fig. 1), the mean $E$ value was about $(97 \pm 3)$ %. The amount of $NO_2^-$ measured on second denuders was

reproducible and equivalent to blanks, representing on average 3 % of the quantity measured on the first denuders. In light of these results, denuders in second position were not subjected to isotopic analysis and allowed trapping efficiency control.

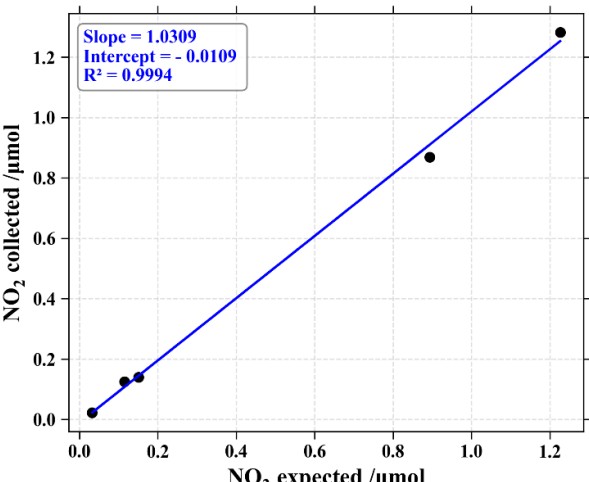

**Figure 1.** Correlation plot of $NO_2$ collected on the first denuder tube of the sampling cartridge vs. $NO_2$ produced by the gas standard generator.

## 2.2 Isotopic analysis

Simultaneous isotopic analyses of $\delta^{15}N$, $\delta^{18}O$, and $\delta^{17}O$ were performed using a Finnigan$^{TM}$-MAT253 isotope ratio mass spectrometer (IRMS) following techniques described by Casciotti et al. (2002) and Kaiser et al. (2007). The azide method (McIlvin and Altabet, 2005) was used with $\approx$ 100 nmol of nitrites converted to $N_2O$ using a 50:50 by volume mixture of 2 M sodium azide and 100 % acetic acid. This chemical method has the advantage over the bacterial method to be free of nitrate interferences since $HNO_3$ is certainly trapped with $NO_2$ in the basic solution coating of the denuder tube. The principle of identical treatment (Brand, 1996) was strictly respected where the standards and samples possessed the same nitrite concentration, water isotopes, total volume and matrix. Three international $KNO_2$ salt standards, RSIL-N7373, RSIL-N10219, and RSIL-N23 with respective $\delta^{15}N/\delta^{18}O$ values of –79.6/4.2 ‰, 2.8/88.5 ‰, and 3.7/11.4 ‰ were used for normalisation of $\delta$-scale. Scale contraction factors were obtained with the linear regression between measured and known values of $\delta^{15}N$ and $\delta^{18}O$. Although the three standards cover a wide range of isotopic composition in $\delta^{15}N$ and $\delta^{18}O$, they do not have an isotopic anomaly in $^{17}O$. For $\delta^{17}O$-scale, MDF fractionation slope (0.52) is assumed for two of these laboratory-prepared nitrite standards (see Appendix A for more details). Accuracy of this analytical method on $\delta^{17}O$, $\delta^{18}O$ and $\delta^{15}N$ measurements was

estimated as the standard deviation ($\sigma$) of the residuals between our measurements of the RSIL standards and their expected values. Additionally, isotopic integrity from denuders extraction to the analysis by IRMS has been investigated and showed no degradation over several weeks (see Appendix B) confirming that this method is suitable for isotopic analysis of $NO_2$, as first demonstrated by Walters et al. (2018). The uncertainties applied to our measurements of $\delta^{15}N$, $\delta^{17}O$ and $\delta^{18}O$ are reported as the propagation error of the measurement uncertainty and the uncertainty resulting from sample storage. Uncertainty on

$\Delta^{17}O$ is derived from the propagation error of the overall uncertainty on $\delta^{17}O$ and $\delta^{18}O$. In our study, average uncertainties on $\delta^{15}N$, $\delta^{17}O$, $\delta^{18}O$, and $\Delta^{17}O$ are estimated to be $\pm 0.1$, $\pm 1.1$, $\pm 2.5$ and $\pm 1.7$ ‰, respectively (1$\sigma$ uncertainties).

### 2.3 Study site and atmospheric NO₂ collection

Atmospheric $NO_2$ was collected at the Université Grenoble Alpes campus site. Located to the eastern Grenoble urban area (690 000 inhabitants), the campus stands between a major transportation route and the Isère river. The city is located at the

confluence of three valleys surrounded by mountain chains that influence the atmospheric dynamics and the local air quality. During winter, persistent temperature inversions combined with intense domestic heating can lead to severe $PM_{10}$ pollution events (Largeron and Staquet, 2016) with daily-average concentration above World Health Organisation thresholds. In summer, emissions are mainly controlled by road traffic that can result in heightened ozone concentrations, especially during stagnant conditions.

Samplings were conducted on a platform five meters above the ground surface. Ambient air was drawn through the cartridge with a Millipore vacuum pump at a flow rate of 10 L min$^{-1}$ (room temperature and one atmospheric pressure) adjusted using a Cole-Palmer$^{TM}$ flowmeter (accuracy $\pm 3$ %). In order to capture the daily variability in $NO_2$ isotopic composition, samples were collected during 24 hours with 3 h sampling intervals during the day, and 5 h sampling from midnight to 5:00 am. Ambient NO and $NO_2$ concentrations were measured with a 2B Technologies$^{TM}$ NO monitor model 410 paired with a $NO_2$

converter model 401.

Honeycomb denuders were cleaned and coated the day before sampling. After being generously rinsed (5 minutes under a stream of deionised water), the denuders were placed in a vacuum chamber (Thermo Scientific$^{TM}$ Refrigerated VaporTrap paired with a SpeedVac Concentrator) and dried at 40 °C during 1 hour. Then, denuders internal walls were individually coated with 10 ml of a 95:5 by volume mixture of 2.5 M KOH (in methanol) and ultrapure guaiacol prepared daily. Denuders were

then drawn off, dried in the vacuum chamber at 40 °C during 30 minutes to minimize blanks, hermetically sealed and stored at ambient temperature in the dark until usage. The different components of the cartridge (impactor, filters, denuders) were cleaned, dried and fitted together just before use. At the end of the sampling period both denuders were removed from the ChemComb cartridge and rinsed with 10 ml of deionised water in order to leach trapped $NO_2$ out. 1 ml of the eluent was rapidly used to determine the nitrite concentration using the Griess-Saltzman reaction and UV-vis spectrometry at 544 nm.

Recovered eluent ($\approx 7$ ml by denuder) was poured in a labelled 15 ml Corning® and stored in a freezer until isotopic analysis the following days.

## 3 Atmospheric observations and multi-isotopic measurements

### 3.1 NO$_x$ and O$_3$ atmospheric observations

Figure 2 shows the time evolution of the hourly NO$_2$, NO and, O$_3$ mixing ratio measured during the period covering two nights
and one day (from 14 May 2019 21:00 to 16 May 2019 5:00). Note that most of our NO measurements are found to be within
the reported detection limit of the instrument except in the morning (see Table 1) and therefore, have to be treated with lot of
caution. NO$_2$ mixing ratios during the sampling period (($6.1 \pm 4.2$) nmol mol$^{-1}$; mean $\pm$ one standard deviation) are in good
agreement with the range of values measured at the local air quality site located a kilometre south of the sampling site
(https://www.atmo-auvergnerhonealpes.fr/).

During both nights, most of the NO$_x$ are in the form of NO$_2$. After sunrise, there is a rapid interconversion between NO and
NO$_2$, driven by NO$_2$ photolysis and reactions of NO with O$_3$ and peroxy radicals (Jacob, 1999). NO$_2$ levels are maximum on
15 May between 4:00 and 10:00 with a sharp peak of 21 nmol mol$^{-1}$ at 8:00. After the morning rise, NO$_2$ decreases to reach a
background concentration of about ($3.0 \pm 0.5$) nmol mol$^{-1}$. This diurnal variation is common in urban/suburban sites
characterised by a morning peak caused by important NO$_x$ emissions, mainly from road traffic (Mayer, 1999). As morning
progresses, the boundary layer height increases rapidly, favouring fast dilution of NO$_x$ concentrations. Moreover, during the
day, NO$_2$ is converted to HNO$_3$, notably by its reaction with OH radicals. Thus, NO$_x$ concentration remains low during the day
likely because of the combination of atmospheric dilution by vertical mixing and efficient chemical conversion by OH and
organic radicals (Tie et al., 2007). In dense urban areas, a second NO$_x$ traffic emission peak can occur in late afternoon but it
is not observed at our sampling site for that specific day. This surface pollution peak is usually weaker than the morning peak
due to an elevated boundary layer and a longer period of evening commuting. After sunset, NO$_2$ concentration increases gently
and reaches a smooth peak with a maximum of 12 nmol mol$^{-1}$ around 1:00 am local time, also recorded at the local air quality
site. This NO$_2$ concentration rise may be due to low NO emissions (converted to NO$_2$ by reaction with O$_3$) combined with a
decreasing boundary layer height during the night which traps atmospheric species close to the surface (Tie et al., 2007; Villena
et al., 2011).

Ozone also exhibits a diurnal variation typical of urban areas (Velasco et al., 2008). O$_3$ peaks around 50 nmol mol$^{-1}$ at the
beginning of both nights to then declines continuously. Indeed, after sunset, O$_3$ production ceases and its concentration drops
due to its dry deposition, reactions with organics, and O$_3$ titration by NO emitted from evening traffic, heating, and industrial
activities in the stable nocturnal boundary layer (Klein et al., 2019). O$_3$ reaches a minimum (about 15 nmol mol$^{-1}$) not at the
end of the night but during the morning rush hours peak of NO. O$_x$ (= O$_3$ + NO$_2$) is a more conservative quantity than O$_3$
because it is less affected by conversion of O$_3$ into NO$_2$ through NO titration which is important in urban environments
(Kleinman et al., 2002). For instance, between 6:00 and 8:00 am, O$_3$ is strongly titrated by freshly emitted NO with its
concentration dropping to about 15 nmol mol$^{-1}$ while O$_x$ reaches a moderate minimum of 34 nmol mol$^{-1}$. After this morning
drop, O$_3$ recovers rapidly to about 46 nmol mol$^{-1}$ in the late morning, possibly caused by downward O$_3$ flux associated with
the formation of the day-time thick boundary layer (Jin and Demerjian, 1993; Klein et al., 2019). During the rest of the day,

O₃ and Oₓ keep increasing gently due to photochemical production and reach a close maxima at the end of the afternoon (Geng et al., 2008). After sunset, the important decline of both O₃ and Oₓ highlights the physical losses, notably O₃ deposition, and chemical loss of NOₓ, typical of urban area.

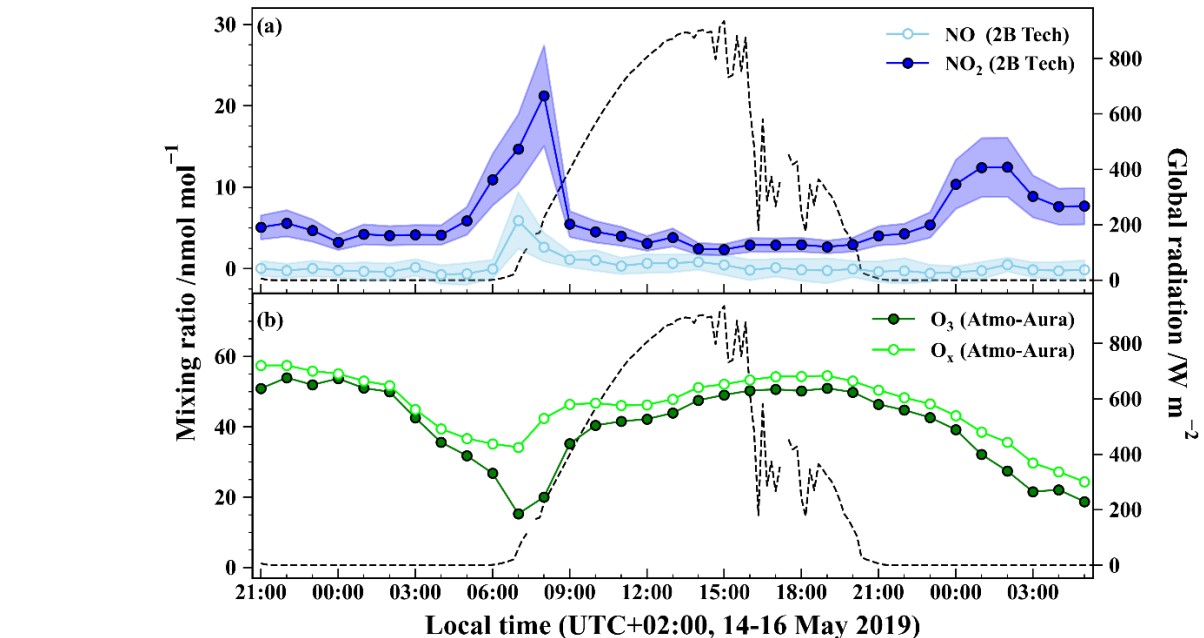

**Figure 2.** Temporal evolution of **(a)** NO (open circles) and NO₂ (close circles) at the sampling site (the envelops represent ± 1σ variations over 1 hour) and of **(b)** O₃ (close circles) and Oₓ (= O₃ + NO₂; open circles) at the air quality station during the sampling period. Markers represent for (a) the hourly mean derived from 1-min measurements and for (b) the hourly mean provided by the air quality station. Global solar radiation flux is represented by dashed lines (measured at 200 meters from the sampling site by the IGE weather station with a Skye SP1110 pyranometer).

## 3.2 Multi-isotopic composition measurements of atmospheric NO₂

We present the data for the multi-isotopic composition of seven atmospheric NO₂ samples while two additional samples were rejected as NO₂⁻ amounts were too low to perform a reliable analysis. Table 1 reports ambient mean concentrations of NO, NO₂ and, O₃ for the isotopic sampling intervals and corresponding measured NO₂ isotopic composition ($\delta^{15}N(NO_2)$, $\delta^{18}O(NO_2)$, and $\Delta^{17}O(NO_2)$). Figure 3 depicts the time series of measured $\delta^{15}N$, $\delta^{18}O$, and $\Delta^{17}O$ of atmospheric NO₂. The temporal evolution of NO₂ oxygen and nitrogen isotopic composition is interpreted in the following section.

| Sampling date & time (start - end) | NO (± 2.5 nmol mol⁻¹) | NO₂ (± 2.5 nmol mol⁻¹) | O₃ [*] (± 6.8 nmol mol⁻¹) | $\delta^{15}N(NO_2)$ (± 0.1 ‰) | $\delta^{18}O(NO_2)$ (± 2.5 ‰) | $\Delta^{17}O(NO_2)$ (± 1.7 ‰) |
|---|---|---|---|---|---|---|
| 14/5/19 21:00 - 00:00 | 0.0 | 5.1 | 52.3 | −11.7 | 75.6 | 27.4 |
| 15/5/19 06:00 - 09:00 | 2.9 | 15.6 | 20.7 | −4.9 | 97.6 | 31.8 |
| 15/5/19 09:00 - 12:00 | 0.8 | 4.7 | 39.1 | −10.1 | 114.5 | 39.2 |
| 15/5/19 12:00 - 15:00 | 0.7 | 3.1 | 44.6 | −11.8 | 90.9 | 35.8 |
| 15/5/19 15:00 - 18:00 | 0.2 | 2.7 | 50.0 | −11.0 | 86.9 | 31.1 |
| 15/5/19 18:00 - 21:00 | 0.0 | 2.9 | 50.3 | −11.1 | 77.1 | 29.7 |
| 16/5/19 00:00 - 05:00 | 0.0 | 9.9 | 26.9 | −11.1 | 62.2 | 20.5 |

**Table 1.** Summary table of sampling periods (dates, local times), NO, NO₂ and O₃ mean mixing ratios over the collection periods, and calibrated NO₂ isotopic measurements of $\delta^{15}N$, $\delta^{18}O$, and $\Delta^{17}O$. All the sampling periods lasted 3 hours except the last one that lasted 5 hours. Averaged measurement uncertainties are provided just below the species names. [*] Data monitored at the local air quality site of Saint-Martin d'Hères located a kilometre south of the sampling site (https://www.atmo-auvergnerhonealpes.fr/).

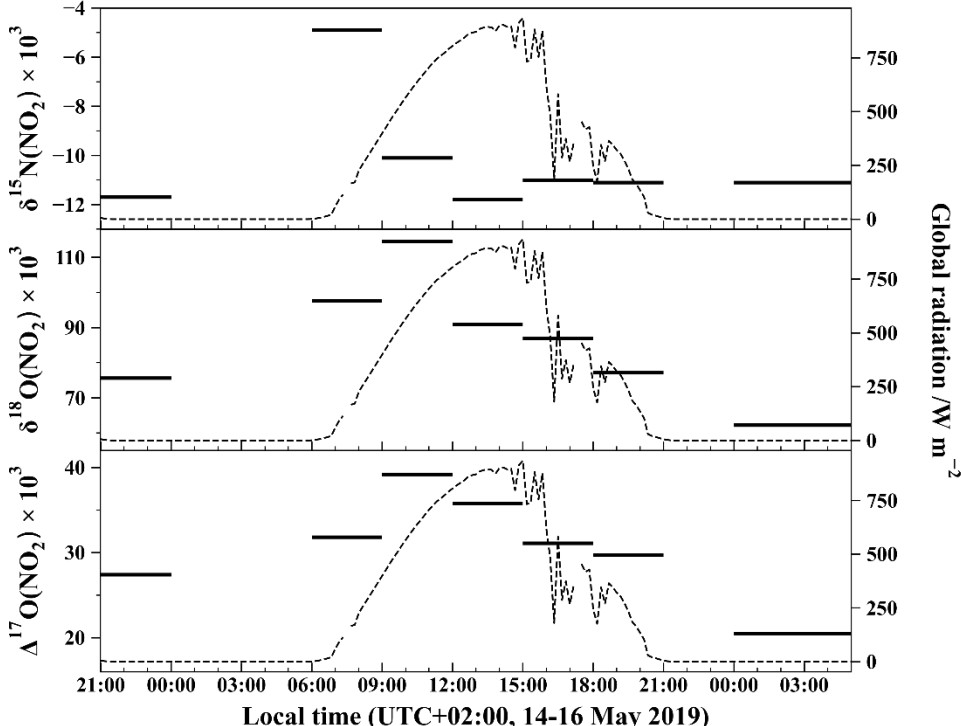

**Figure 3.** Temporal evolution of $\delta^{15}N$, $\delta^{18}O$ and, $\Delta^{17}O$ of atmospheric NO₂ measured with the azide method. Isotopic values for each 3 hours slots are from the same NO₂ sample collected over 3 hours (except for the last period which lasted 5 hours). Global solar radiation flux is represented by dashed lines (measured at 200 meters from the sampling site by the IGE weather station with a Skye SP1110 pyranometer).

## 4 Discussion of the multi-isotopic composition of atmospheric NO₂

### 4.1 Oxygen isotope composition

The time evolution of $\delta^{18}O$ of atmospheric $NO_2$ ($\delta^{18}O(NO_2)$) shown in Fig. 3 exhibits a substantial diurnal variation with a day mean of $(93.4 \pm 13.9)$ ‰ and a night mean of $(68.9 \pm 9.5)$ ‰. A maximum value of 114.5 ‰ is observed in the morning (09:00-12:00 interval) and a minimum value of 62.2 ‰ for the late-night interval (00:00-05:00). Using a similar sampling apparatus during summer in the urban/sub-urban site of West Lafayette, USA, Walters et al. (2018) reported $\delta^{18}O(NO_2)$ daytime and nighttime mean values of $(86.5 \pm 14.1)$ ‰ and $(56.3 \pm 7.1)$ ‰, respectively. Although our daytime values are higher than those of Walters et al. (2018), both datasets exhibit the same day-night contrast with a maximum during the day and a minimum at night. As expected from $\delta^{18}O$ values, $\Delta^{17}O(NO_2)$ follows a similar diurnal variation with a maximum value of 39.2 ‰ for the 09:00-12:00 interval and a minimum value of 20.5 ‰ for the 00:00-05:00 interval. High $\Delta^{17}O$ values are expected to reflect the importance of ozone in the oxidation of NO to $NO_2$. Since daytime and nighttime chemistries are radically different, interpretations of our $\Delta^{17}O$ measurements and their implications are discussed separately by day and night.

### 4.1.1 Fundamentals of NO$_x$ chemistry and isotopic transfers

$NO_x$ are mainly produced under the form of NO by combustion and lighting processes (Dennison et al., 2006; Young, 2002) and by the biological activity of soils (Davidson and Kingerlee, 1997). In the daytime, NO and $NO_2$ rapidly interconvert in a time scale of about 1-2 minutes establishing a photostationary steady state (PSS; Leighton 1961):

$$NO_2 + h\nu \overset{M}{\rightarrow} O(^3P) + NO \tag{R1}$$

$$O(^3P) + O_2 \overset{M}{\rightarrow} O_3 \text{ with } M = N_2 \text{ or } O_2 \tag{R2}$$

$$NO + O_3 \rightarrow NO_2 + O_2 \tag{R3}$$

This so-called null cycle can be disturbed by $RO_2$ radicals when $NO_x$ concentrations are relatively high, typically above 30 pmol mol$^{-1}$ (Seinfeld and Pandis, 2006):

$$NO + RO_2 \rightarrow NO_2 + RO \tag{R4}$$

The reaction between NO and $RO_2$ competes with the NO + $O_3$ reaction, allowing $NO_2$ formation without the consumption of an ozone molecule in the cycle (Monks, 2005). This results in ozone production and can lead to severe ozone build up in polluted areas. At night, $RO_2$ concentrations are strongly reduced making ozone the main NO oxidant following R3.

$NO_x$ are mainly removed from the atmosphere via the oxidation of $NO_2$ into nitric acid during the day:

$$NO_2 + OH \overset{M}{\rightarrow} HNO_3 \tag{R5}$$

and at night:

$$270 \quad NO_2 + O_3 \xrightarrow{M} NO_3 + O_2 \tag{R6}$$

$$NO_3 + NO_2 \xrightarrow{M} N_2O_5 \xrightarrow{H_2O, \ aerosol} 2 \ NHO_3 \tag{R7}$$

In this framework, $\Delta^{17}O(NO_2)$ is driven by the relative importance of the different $NO_2$ production channels because $NO_2$ loss processes do not fractionate in terms of oxygen mass-independent anomaly. Each $NO_2$ production channel generates a specific mass-independent isotopic anomaly $\Delta^{17}O$ on the produced $NO_2$ (Kaiser et al., 2004). Based on the $NO_2$ continuity equation, 275 this can be expressed with the following $\Delta^{17}O(NO_2)$ mass-balance equation (Morin et al., 2011):

$$\frac{d}{dt}\left([NO_2] \times \Delta^{17}O(NO_2)\right) = \sum_i \left(P_i \times \Delta^{17}O_i(NO_2)\right) - \left(\sum_j L_j\right) \times \Delta^{17}O(NO_2) \tag{2}$$

with $[NO_2]$ being the atmospheric $NO_2$ concentration, $P_i$ and $L_j$ the $NO_2$ production/emission and loss rates (= concentration of involved species multiplied by the kinetic constant of the considered chemical reaction), and $\Delta^{17}O_i(NO_2)$ the specific isotope anomaly transferred to $NO_2$ through the production reaction $i$.

**4.1.2 $\Delta^{17}O_{day}(NO_2)$**

By day, the $NO_x$ photochemical cycle (R1 to R4) achieves a steady state in 1-2 minutes, which is several orders of magnitude faster than $NO_2$ loss reactions (Atkinson et al., 1997) and emission rate ($NO_x$ are mainly emitted under the form of NO; Villena et al., 2011). It follows that NO and $NO_2$ short variations can be neglected i.e. $\frac{d}{dt}[NO_2] \approx 0$ and $\frac{d}{dt}[NO] \approx 0$ on short timescales. In addition, fast interconversions between NO and $NO_2$ generate quickly an isotopic equilibrium between NO and $NO_2$ 285 resulting in $\Delta^{17}O(NO_2) \approx \Delta^{17}O(NO)$ (Michalski et al., 2014; Morin et al., 2007). With these approximations, considering only the main reactions and neglecting halogen chemistry, Eq.(2) yields to (Morin et al., 2007):

$$\Delta^{17}O_{day}(NO_2) \approx \frac{k_{NO+O_3}[O_3] \times \Delta^{17}O_{NO+O_3}(NO_2) + k_{NO+RO_2}[RO_2] \times \Delta^{17}O_{NO+RO_2}(NO_2)}{k_{NO+O_3}[O_3] + k_{NO+RO_2}[RO_2]} \tag{3}$$

with $\Delta^{17}O_{NO+O_3}(NO_2)$ being the ozone isotopic anomaly transferred to NO during its oxidation to $NO_2$ via R3 (also called the transfer function of the isotope anomaly of ozone to $NO_2$; Savarino et al., 2008) and $\Delta^{17}O_{NO+RO_2}(NO_2)$ being the $RO_2$ isotopic 290 anomaly transferred to NO during its oxidation to $NO_2$ via R4. $\Delta^{17}O_{NO+RO_2}(NO_2)$ can be considered to be negligible (Alexander et al., 2020; Michalski et al., 2003) because $RO_2$ are mainly formed by the reactions $R + O_2$ and $H + O_2$ and the isotopic anomaly of atmospheric $O_2$ is very close to 0 ‰ (Barkan and Luz, 2003). This assumption has been estimated to affect the overall $\Delta^{17}O$ of $RO_2$ values by less than 1 ‰ (Röckmann et al., 2001). As a result, Eq.(3) can be simplified, giving a $\Delta^{17}O_{day}(NO_2)$ driven by the relative importance of R3 ($NO + O_3$) and R4 ($NO + RO_2$) in the NO oxidation and by the oxygen 295 isotopic anomaly transferred from $O_3$ to $NO_2$:

$$\Delta^{17}O_{day}(NO_2) \approx T_{NO+O_3} \times \Delta^{17}O_{NO+O_3}(NO_2) \tag{4}$$

with $T_{\mathrm{NO+O_3}} = \dfrac{k_{\mathrm{NO+O_3}}[\mathrm{O_3}]}{k_{\mathrm{NO+O_3}}[\mathrm{O_3}] + k_{\mathrm{NO+RO_2}}[\mathrm{RO_2}]}$ (5)

$\Delta^{17}O_{\mathrm{NO+O3}}(\mathrm{NO_2})$ has been determined experimentally by Savarino et al. (2008). They reported $\Delta^{17}O_{\mathrm{NO+O_3}}(\mathrm{NO_2}) = (1.18 \pm 0.07 \times \Delta^{17}O(\mathrm{O_3}) + 6.6 \pm 1.5)$ with $\Delta^{17}O(\mathrm{O_3})$ being the bulk ozone isotopic anomaly. $\Delta^{17}O(\mathrm{O_3})$ has been measured in Grenoble in 2012 (Vicars and Savarino, 2014) with a mean value of $(26.2 \pm 1.3)$ ‰, corresponding to a $\Delta^{17}O_{\mathrm{NO+O_3}}(\mathrm{NO_2})$ value of $(37.5 \pm 2.8)$ ‰ which, according to Eq.(4), would give a maximum $\Delta^{17}O_{\mathrm{day}}(\mathrm{NO_2})$ value of $(37.5 \pm 2.8)$ ‰. It is consistent with our maximum measured $\Delta^{17}O(\mathrm{NO_2})$ value of 39.2 ‰ for the 09:00-12:00 interval. In light of the known uncertainties, the small difference is not significant and is much smaller than the diurnal variations of $\Delta^{17}O(\mathrm{NO_2})$. Note that the $\Delta^{17}O$ calibration is not very accurate for the most enriched samples because nitrite standards with high $\Delta^{17}O$ are still not readily available. In a laboratory study Michalski et al. (2014) measured the $\Delta^{17}O$ of $\mathrm{NO_2}$ formed by the photochemical NO-$\mathrm{NO_2}$-$\mathrm{O_3}$ cycle and reported $\Delta^{17}O(\mathrm{NO_2}) = (39.3 \pm 1.9)$ ‰. Despite experimental conditions that are not strictly applicable to our atmospheric conditions (e.g. $\mathrm{NO_x} \gg \mathrm{O_3}$, light source, absence of VOCs), their value is surprisingly close to our maximum value. Assuming that our maximum $\Delta^{17}O(\mathrm{NO_2})$ value correspond to $T_{\mathrm{NO+O_3}}$ close to unity (R3 ($\mathrm{NO + O_3}$) $\gg$ R4 ($\mathrm{NO + RO_2}$)), we use a value of 39.2 ‰ for $\Delta^{17}O_{\mathrm{NO+O_3}}(\mathrm{NO_2})$ for the following calculations. Combining Eq.(4) and Eq.(5), an expression for the $\mathrm{RO_2}$ concentration can be derived as:

$$[\mathrm{RO_2}] = \frac{k_{\mathrm{NO+O_3}}[\mathrm{O_3}]}{k_{\mathrm{NO+RO_2}}}\left(\frac{\Delta^{17}O_{\mathrm{NO+O3}}(\mathrm{NO_2})}{\Delta^{17}O_{\mathrm{day}}(\mathrm{NO_2})} - 1\right)$$ (6)

Figure 4 shows the daytime evolution of $T_{\mathrm{NO+O_3}}$ calculated from Eq.(4) and $\mathrm{RO_2}$ calculated from Eq.(6). $T_{\mathrm{NO+O_3}}$ varies between 0.76 and 1 with a mean daytime of 0.86 (the measured daytime $\Delta^{17}O(\mathrm{NO_2})$ mean value is $(33.5 \pm 3.9)$ ‰) meaning that 86 % of $\mathrm{NO_2}$ is formed via R3 (oxidation of NO by $\mathrm{O_3}$). The mean estimated $\mathrm{RO_2}$ concentration is $(13.8 \pm 11.2)$ pmol mol$^{-1}$. Note that $\mathrm{RO_2} = 0$ pmol mol$^{-1}$ for the 09:00-12:00 interval originates from our assumption of $T_{\mathrm{NO+O_3}} = 1$ for our highest $\Delta^{17}O(\mathrm{NO_2})$ value; in reality, it only means that $\mathrm{RO_2}$ is so low that R3 ($\mathrm{NO + O_3}$) $\gg$ R4 ($\mathrm{NO + RO_2}$). Overall, our $\mathrm{RO_2}$ values are found to be within the range of values measured at urban/peri-urban sites (see Table 2). However, $\mathrm{RO_2}$ diurnal variation at our site does not follow the pattern of previous measurements which usually report a diurnal variation with a maximum varying from noon to early afternoon (Fuchs et al., 2008; Tan et al., 2017) whereas this study shows a maximal concentration in late afternoon. With such a limited dataset (only 1 day of sampling), it is not possible to draw general conclusions on the $\mathrm{NO_x}/\mathrm{RO_2}$ chemistry dynamics. An important recommendation for further investigation is to conduct isotopic measurements with accurate measurements of key atmospheric radicals/oxidants, e.g. NO, $\mathrm{O_3}$, and possibly $\mathrm{RO_2}$, in order to test quantitatively our isotopic approach. Additionally, the use of a chemical box-model is also recommended because it would allow to account for non-equilibrium effects in isotopic transfers and thus strengthen the interpretation of isotopic measurements in the investigation of the reactive nitrogen cycle in urban atmospheres.

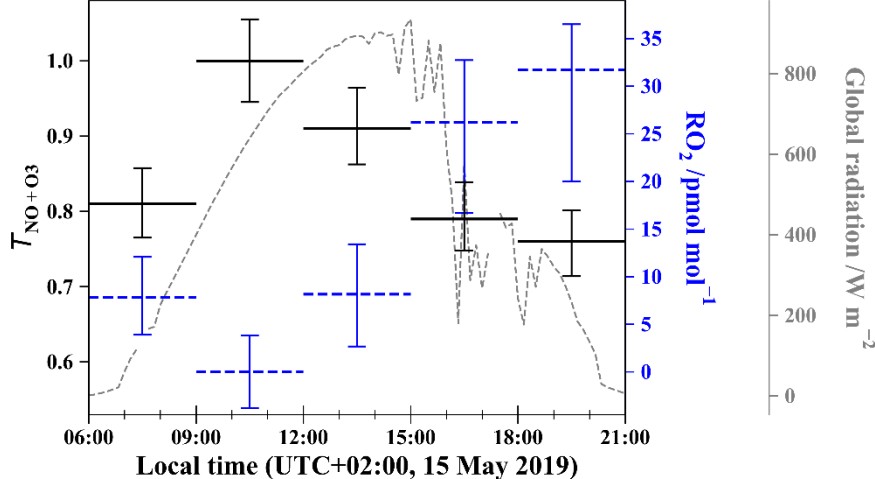

**Figure 4.** Daytime evolution $T_{NO+O_3}$ (solid black slots) estimated from Eq.(4) using measured $\Delta^{17}O(NO_2)$ in Grenoble, and of $RO_2$ concentrations (dashed blue slots) estimated from Eq.(6). Error bars for $T_{NO+O_3}$ are derived from standard deviations of $\Delta^{17}O(NO_2)$ and $\Delta^{17}O(O_3^*)$ measured in Grenoble (Vicars and Savarino, 2014). $RO_2$ error bars are derived from $O_3$ measurement uncertainties and errors on $T_{NO+O_3}$ (by comparison, errors on reaction constants can be neglected). Global solar radiation flux is represented by dashed lines (measured at 200 meters from the sampling site by the IGE weather station with a Skye SP1110 pyranometer).

| Site | $RO_2$ /pmol mol$^{-1}$ | Reference |
|---|---|---|
| Grenoble (2019, May) | 0-35 [*] | This study |
| UK, suburban site (2003, July-August) | 4-22 | Emmerson et al. (2007) |
| Germany, suburban site (2005, July) | 2-40 | Fuchs et al. (2008) |
| Germany, rural site (1998, July-August) | 2-50 | Mihelcic et al. (2003) |
| USA, rural site (2002, May-June) | 9-15 | Ren et al. (2005) |
| China, rural site (2014, June-July) | 7-37 | Tan et al. (2017) |

**Table 2.** Mean daytime $RO_2$ concentration ranges measured during field campaigns in various environments and seasons. [*]Derived from Eq.(6) using $\Delta^{17}O$ values of atmospheric $NO_2$ in Grenoble.

Morin et al. (2011) simulated the diurnal variation of $\Delta^{17}O(NO_2)$ in a remote marine boundary layer without the effect of $NO_x$ emissions. They assumed $\Delta^{17}O(O_3) = 30$ ‰ ($\Delta^{17}O_{NO+O_3}(NO_2) = 45$ ‰) resulting into higher overall $\Delta^{17}O(NO_2)$ values compared to our study. Their simulated $\Delta^{17}O(NO_2)$ exhibited large diurnal variations with maximum values at night (close to 41 ‰) and minimum values at noon of 28 ‰. This is consistent with $RO_2$ concentration reaching a maximum around local

noon in clean environments. In contrast to their model simulations, our daytime $\Delta^{17}O(NO_2)$ measurements are higher than our nighttime measurements. We will show in the following section that this difference originates from absence of $NO_x$ emissions in Morin et al. (2011) photochemical modelling.

### 4.1.3 $\Delta^{17}O_{night}(NO_2)$

Without photolysis at night and associated $RO_2$ production, ozone is the unique NO oxidant. NO and $NO_2$ are no longer in photochemical equilibrium because $NO_2$ cannot be converted back into NO. As a result, the oxygen isotopic composition of $NO_2$ formed during the night is determined by the oxygen isotopic composition of $O_3$ and emitted NO. Additionally, in order to estimate the overall isotopic signature of sampled $NO_2$ at night, we need to determine the residuals of $NO_2$ formed during the day that is still present during the night, following:

$$\Delta^{17}O_{night}(NO_2) \approx x \times \Delta^{17}O_{day}(NO_2) + \frac{(1-x)}{2} \times (\Delta^{17}O_{NO+O_3}(NO_2) + \Delta^{17}O(NO)) \tag{7}$$

with $x$ being the fraction of $NO_2$ formed during the day to the total $NO_2$ measured at night and $(1-x)$ representing the fraction of $NO_2$ which has been produced during the night to the total $NO_2$ measured at night. NO is mainly emitted by combustion processes in which a nitrogen atom (from atmospheric $N_2$ or N present in fuel) is added to an oxygen atom formed by the thermal decomposition of $O_2$ (Zeldovich, 1946). With $\Delta^{17}O(O_2)$ being close to 0 ‰ (Barkan and Luz, 2003), NO emissions are very likely to have a $\Delta^{17}O \approx 0$ ‰, or at least negligible compared to $\Delta^{17}O_{NO+O_3}(NO_2)$. Using Eq.(7) and assuming a negligible isotope anomaly for NO, the time evolution of $\Delta^{17}O(NO_2)$ over the night can be calculated. It is worth pointing out that the $x$ fraction becomes very small at the end of the night allowing to further simplify Eq.(7): $\Delta^{17}O_{night}(NO_2) = \frac{1}{2} \times \Delta^{17}O_{NO+O_3}(NO_2)$. Thus, if there are nighttime NO emissions, a measurement of $\Delta^{17}O(NO_2)$ at the end of the night is also an interesting way of deriving $\Delta^{17}O(O_3)$ which is difficult to measure directly. The nighttime variation of the $x$ fraction is estimated considering that the nighttime lifetime of $NO_2$ relative to oxidation via ozone and dry deposition is 7.2 hours ($O_3$ chemical sink is dominant over deposition by a factor $> 10^4$ with $k_{NO_2+O_3} = 1.4 \times 10^{-13}$ exp[−2470/T] cm$^3$ molecule$^{-1}$ s$^{-1}$ Atkinson et al., 2004; $NO_2$ dry velocity $V_d = 0.25$ cm s$^{-1}$ Holland et al., 1999 and assuming a nighttime boundary layer height of 500 m). For the 00:00-05:00 interval, we calculate a mean value of $\Delta^{17}O(NO_2) = 19.9$‰ (with an overall error of about 1.6 ‰) which is very close to our measured $\Delta^{17}O(NO_2)$ of 20.5 ‰. This first dataset of nighttime $\Delta^{17}O(NO_2)$ measurements appears to confirm our understanding of nocturnal $NO_2$ formation (Alexander et al., 2020; Michalski et al., 2014). NO emissions in urban areas have a very significant influence on $\Delta^{17}O(NO_2)$ leading to a behaviour in opposition to the one observed in remote locations. As illustrated by Morin et al. (2011), $\Delta^{17}O(NO_2)$ is predicted to be maximal at night in remote areas where NO emissions are negligible, reflecting the isotopic signature of $NO_2$ at sunset. In areas where nighttime NO emissions are high, nighttime $\Delta^{17}O(NO_2)$ can be up to a factor of two smaller than in remote areas.

## 4.2 Nitrogen isotope composition

Measured $\delta^{15}N(NO_2)$ values range from $-11.8$ to $-4.9$ ‰ with no clear diurnal variation and values clustering around an overall mean of $(-10.2 \pm 2.2)$ ‰ (see Fig. 3). Using a similar method, Walters et al. (2018) collected atmospheric $NO_2$ over one month in a urban/sub-urban location during the summer. They reported a mean $\delta^{15}N$ value of $(-11.4 \pm 6.9)$ ‰, very close to our mean value but with a wider overall range (from $-31.4$ to $+0.4$ ‰). In another urban area but using passive samplers, Dahal and Hastings (2016) reported mean $\delta^{15}N(NO_2)$ values of $(-8.3 \pm 0.9)$ ‰ and $(-6.4 \pm 1.4)$ ‰ for summer and winter periods, respectively. All these values are within the $\delta^{15}N$ range for NO emitted by industrial combustion and traffic sources which are reported to vary from $-19.7$ to $-13.7$ ‰ and from $-9$ to $-2$ ‰ respectively (Miller et al., 2017; Walters et al., 2015). Interestingly, all the $\delta^{15}N$ values measured at our sampling site fall within a narrow range, from about $-12$ to $-10$ ‰, except for the sample collected between 6:00 and 9:00 which has a much higher value of $-4.9$ ‰. This singular value is well correlated with the morning NO traffic emission spike (see Fig. 2). However, once emitted into the atmosphere, NO can undergo isotopic fractionations that modify the nitrogen isotope distribution in $NO_2$ relative to emitted NO (Freyer et al., 1993). In order to use $\delta^{15}N(NO_2)$ as a tracer of $NO_x$ sources, we need to quantify these nitrogen isotopic shifts to correct measured $\delta^{15}N(NO_2)$. Nitrogen isotopic fractionation, defined as $\Delta(NO_2 - NO_x) = \delta^{15}N(NO_2) - \delta^{15}N(NO_x)$, is the result of a combination of three effects: 1) an Equilibrium Isotope Effect (EIE) between NO and $NO_2$ and 2) a Kinetic Isotope Effect (KIE) during NO oxidation to $NO_2$ and 3) a Photochemical isotope fractionation effect (PHIFE) during $NO_2$ photolysis (other $NO_2$ sinks are negligible during the day). The overall daytime nitrogen isotopic shift of $NO_2$ relative to emitted $NO_x$ ($\Delta_{day}(NO_2 - NO_x)$) can be estimated using the steady-state isotopic mass balance for $NO_2$. Li et al. (2020) derived an expression for $\Delta(NO_2 - NO_x)$ assuming that the conversion of NO to $NO_2$ is solely driven by $O_3$. This could therefore lead to uncertainties on the $NO_2$ shift when other conversion pathways become significant with respect to the NO conversion by $O_3$. A more general expression for $\Delta(NO_2 - NO_x)$ can be derived taking into account the conversion of NO to $NO_2$ via other species, notably $RO_2$ (see equation C11 in appendix C). In our urban environment, we only consider the conversion of NO into $NO_2$ via $O_3$ and $RO_2$ during the day. Assuming $\alpha_{KIE(NO+O_3)} \approx \alpha_{KIE(NO+RO_2)}$ (see derivation in appendix C), $\Delta_{day}(NO_2 - NO_x)$ can be expressed by

$$\Delta_{day}(NO_2 - NO_x) = \frac{\alpha_{LCIE}{}^* A^*{}_{day} + (\alpha_{EIE} - 1)}{A^*{}_{day} + 1}\left(1 - f_{NO_2}\right) \tag{8}$$

with $\alpha_{LCIE}{}^* = \alpha_{KIE(NO+O_3)} - \alpha_{PHIFE}$

and $A^*{}_{day} = \frac{J_{NO_2}}{k_{NO+NO_2}[NO]} = \frac{k_{NO+O_3}[O_3] + k_{NO+RO_2}[RO_2]}{k_{NO+NO_2}[NO_2]}$

where $f_{NO_2} = [NO_2]/[NO_x]$, $\alpha_{LCIE}{}^*$ the fractionation factor of combined KIE and PHIFE and $\alpha_{EIE}$ the EIE fractionation factor. $A^*{}_{day}$ is defined as the ratio of the $NO_2$ lifetime with respect to isotopic exchanges over the daytime $NO_2$ chemistry lifetime. $J_{NO_2}$ is the $NO_2$ photolysis rate, $k_{NO+O_3}$ is the rate constant of reaction $NO + O_3$, $k_{NO+RO_2}$ is the rate constant of reaction NO

+ RO$_2$ and $k_{NO+NO_2}$ is the rate constant of the isotopic exchange (CR1) (see Appendix D for rate constants data). Interestingly, we can combine Eq.(8) and Eq.(6), and express $A^*_{day}$ as a function of oxygen isotopic variables discussed in the previous section:

$$A^*_{day} = \frac{k_{NO+O_3}[O_3]}{k_{NO+NO_2}[NO_2]}\left(\frac{\Delta^{17}O_{NO+O_3}(NO_2)}{\Delta^{17}O_{day}(NO_2)}\right) \tag{9}$$

Since our NO measurements are not precise and we do not have direct measurements of $J_{NO_2}$ or RO$_2$, we use Eq.(9) to estimate the NO$_2$ isotopic fractionation shift. Note that, although Li et al. (2020) only consider the NO conversion via O$_3$ in the analysis of their nitrogen isotopic data, they found an excellent agreement between their calculated values and field isotopic measurements at Jülich, Germany (Freyer et al., 1993). Nonetheless, the reason of this accordance remains unclear, as it could be attributable to an equivalent KIE of NO + O$_3$ and NO + RO$_2$ but also to the dominance of the NO oxidation via O$_3$ over

RO$_2$.

At night, the isotopic fractionation shift $\Delta_{night}(NO_2 - NO_x)$ is driven by EIE, KIE, the N isotopic composition of NO emissions, and $f_{NO_2}$, given that $J_{NO_2}$ is null (see derivation in appendix C):

$$\Delta_{night}(NO_2 - NO_x) \approx \frac{A^*_{night}\left(\alpha_{KIE} - \left(\frac{1+\delta^{15}N(NO_{emis})}{1+\delta^{15}N(NO_2)}\right)\right) + (\alpha_{EIE} - 1)}{A^*_{night} + 1}\left(1 - f_{NO_2}\right) \tag{10}$$

where $\delta^{15}N(NO_{emis})$ is the N isotopic composition of NO emissions

and $A^*_{night} = \frac{k_{NO+O_3}[O_3]}{k_{NO+NO_2}[NO_2]} = \frac{E(NO)}{k_{NO+NO_2}[NO][NO_2]}$                      (11)

with $E(NO)$ the NO emission flux. From laboratory experiments, Li et al. (2020) reported fractionation factors of 1.0289 ± 0.0019 and 0.990 ± 0.005, for $\alpha_{EIE}$ and $\alpha_{LCIE}^*$, respectively. Using these experimental values and the ambient concentrations of ozone, NO and, NO$_2$ measured at our sampling site, we estimate the time evolution of $\Delta(NO_2 - NO_x)$ from Eq.(8) and Eq.(9) for daytime. At night, [NO] << [NO$_2$] and hence $f_{NO_2}$ tends towards 1 and $\Delta_{night}(NO_2 - NO_x) \approx 0$ (Table 3 provides the

calculated values). $\Delta(NO_2 - NO_x)$ values are found to be negligible during the entire sampling period except between 6:00 and 9:00 with a mean $\Delta(NO_2 - NO_x)$ value of 2.7 ‰ due to lower $f_{NO_2}$ and $A^*_{day}$ values. Overall, in our moderately polluted environment, nitrogen oxide isotope effects appear to induce very small nitrogen isotopic shift, reflecting the fact that NO$_x$ is overwhelmingly under the form of NO$_2$ (mean $f_{NO_2}$ = 0.93). Our results are in good agreement with the $\Delta(NO_2 - NO_x)$ range (between 1.3 and 2.5 ‰) calculated from isotopic measurements at West Lafayette, USA (Walters et al., 2018). Moreover, Li

et al. (2020) calculated a mean $\Delta(NO_2 - NO_x)$ of (1.3 ± 3.2) ‰ from isotopic measurements near San Diego, USA (NO$_x$ concentration varied from 1 to 9 nmol mol$^{-1}$).

| Sampling date & time (start - end) | $f_{\mathrm{NO_2}}$ | $A^*$ | $\Delta(\mathrm{NO_2 - NO_x})$ /‰ | $\delta^{15}\mathrm{N(NO_x)}$ /‰ |
|---|---|---|---|---|
| 14/5/19 21:00 - 00:00 | 1.00 | 1.70 | 0.00 | −11.70 |
| 15/5/19 06:00 - 09:00 | 0.87 | 0.27 | 2.72 | −7.62 |
| 15/5/19 09:00 - 12:00 | 0.85 | 1.46 | 0.85 | −10.95 |
| 15/5/19 12:00 - 15:00 | 0.81 | 2.61 | 0.01 | −11.81 |
| 15/5/19 15:00 - 18:00 | 0.95 | 3.38 | −0.13 | −10.87 |
| 15/5/19 18:00 - 21:00 | 1.00 | 3.04 | 0.00 | −11.22 |
| 16/5/19 00:00 - 05:00 | 1.00 | 0.42 | 0.00 | −11.10 |

**Table 3.** Summary of measured $f_{\mathrm{NO_2}}$, calculated $A^*$ values using Eq.(9) for daytime and Eq.(11) for nighttime, calculated isotopic fractionation between $\mathrm{NO_2}$ and $\mathrm{NO_x}$ ($\Delta(\mathrm{NO_2 - NO_x})$) using Eq.(8) for daytime, and Eq.(10) for nighttime and, $\delta^{15}\mathrm{N(NO_x)}$ estimated with $\Delta(\mathrm{NO_2 - NO_x})$ and measured $\delta^{15}\mathrm{N(NO_2)}$.

Using estimated $\delta^{15}\mathrm{N(NO_x)}$, we evaluate the relative contributions of the dominant $\mathrm{NO_x}$ sources at our site using the Bayesian
isotopic mixing model SIAR (Stable Isotope Analysis in R; Parnell et al., 2010). Initially developed for ecological studies (Inger et al., 2006; Samelius et al., 2007), isotopic mixing models have been recently used for atmospheric applications, notably to identify major $\mathrm{NO_x}$ sources of aerosol nitrate from $\delta^{15}\mathrm{N}$ (Jin et al., 2021; Zong et al., 2017; Fan et al., 2019). Using as inputs not only isotopic measurements but also their uncertainties, the SIAR model can be used to calculate potential $\mathrm{NO_x}$ sources solutions as probability distributions. A recent emission inventory of $\mathrm{NO_x}$ in the Grenoble urban area estimated that, in 2016,
52 % of emitted $\mathrm{NO_x}$ could be attributed to transport, 26 % to industries, 20% to the residential/tertiary sectors and, 2 % to agriculture (Topin et al., 2019). Looking at the type of energy consumed by each sector, we estimate that at this time of the year, our sampling site was mostly influenced by fossil-fuel combustion $\mathrm{NO_x}$ sources, mainly gasoline/diesel and natural-gas, and by biogenic $\mathrm{NO_x}$ sources (soils emissions). As shown by previous studies, $\delta^{15}\mathrm{N}$ of $\mathrm{NO_x}$ emitted by vehicles exhausts depends on the fuel type, the reduction emission technology, and the vehicle run time with values ranging from −21 ‰ to −2
‰ (Walters et al., 2015). As 90 % of traffic-$\mathrm{NO_x}$ are emitted by diesel-powered engines in the Grenoble urban area (Atmo-Auvergne-Rhônes-Alpes, 2018), we use a value of (−4.7 ± 1.7) ‰, representative of the U.S. vehicle fleet (Miller et al., 2017) for which about 80 % of its traffic-$\mathrm{NO_x}$ emissions originate from diesel vehicles (Dallmann et al., 2013). For $\delta^{15}\mathrm{N}$ of $\mathrm{NO_x}$ emitted by natural gas combustion, we use a value of (−16.5 ± 1.7) ‰ which is the average isotopic signature of natural gas-burning power plants and residential furnace exhausts (Walters et al., 2015). Despite the large range of $\delta^{15}\mathrm{N}$ values for biogenic
$\mathrm{NO_x}$, (from −59.8 to −19.9 %) (Li and Wang, 2008; Yu and Elliott, 2017; Walters et al., 2015), these values are still very distinct from $\delta^{15}\mathrm{N}$ of fossil-fuel combustion $\mathrm{NO_x}$, making possible to roughly estimate the relative contributions of different $\mathrm{NO_x}$ sources at our sampling site. We use a soil-$\mathrm{NO_x}$ $\delta^{15}\mathrm{N}$ value of (−33.8 ± 12.2) ‰ (Zong et al., 2017) which is the average of values taken from several studies on $\mathrm{NO_x}$ emitted by natural and fertilized soil (Felix and Elliott, 2014; Li and Wang, 2008). Over our sampling period, the SIAR model results indicate traffic as the dominant $\mathrm{NO_x}$ emission source with a mean relative
contribution of (57± 8) % (see Fig. 5). Natural gas combustion is found to be the second main $\mathrm{NO_x}$ emission source (36 ± 12)

% before soil emissions which accounts for only (7 ± 5) %. The limited nature of our measurements dataset (only one day of sampling) prevents us to draw any general and robust conclusions on the relative contributions of $NO_x$ emissions at our site. Nonetheless, we note that the SIAR overall source apportionment is in close agreement with the Grenoble urban area emission inventory concerning traffic emissions (52 % in 2016), lending some support to the idea that $\delta^{15}N$ of $NO_2$ is a reliable tracer of $NO_x$ emission sources after correction for LCIE and EIE.


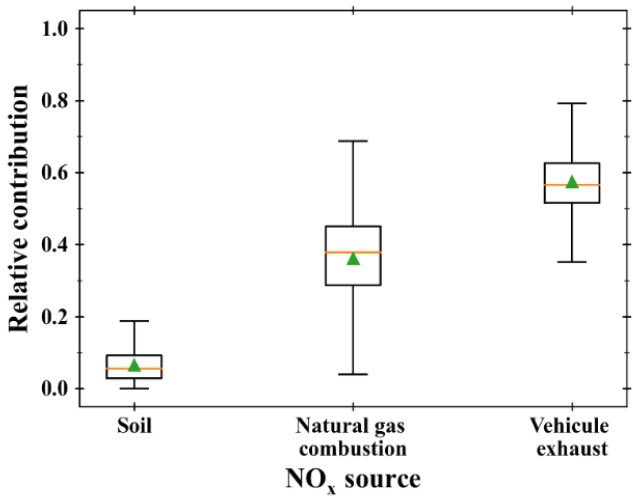

**Figure 5.** Potential $NO_x$ emission source partitioning using the SIAR model based on estimated $\delta^{15}N(NO_x)$. References values for each source were taken from Miller et al. (2017), Walters et al. (2015) and, Zong et al. (2017).

## 5 Conclusion

The primary goal of this preliminary work was to address an efficient and portable sampling system for atmospheric $NO_2$ fitting with accurate isotopic analysis of double nitrogen and triple oxygen isotopes. First simultaneous measurements of the

multi-isotopic composition ($\delta^{15}N$, $\delta^{18}O$, and $\Delta^{17}O$) of atmospheric $NO_2$ are reported here, notably at relatively high temporal resolution (3 h). Over the course of more than one day in the Grenoble urban/suburban environment, $\Delta^{17}O(NO_2)$ is found to vary diurnally with a maximum value of (39.2 ± 1.7) ‰ during the day and a minimum value of (20.5 ± 1.7) ‰ at night. At photo-stationary state, high $\Delta^{17}O(NO_2)$ values result from the ozone predominance in NO oxidation pathways whereas lower values reflect the influence of peroxy radicals. We estimate from our $\Delta^{17}O(NO_2)$ measurements that 86 % of $NO_2$ produced by

day originates from the oxidation of NO by $O_3$. Moreover, a mean daytime peroxy radical concentration of (13.8 ± 11.2) pmol mol$^{-1}$ is derived from the oxygen isotopic measurements. At night, $NO_x$ photochemistry shutdowns and hence $\Delta^{17}O(NO_2)$ decreases under the growing influence of the isotopic footprint from NO emitted by night. The $\Delta^{17}O(NO_2)$ measurement

towards the end of the night is found to be quantitatively consistent with typical values of $\Delta^{17}O(O_3)$. The $\delta^{15}N(NO_2)$ measurements show little variations, from −11.8 to −4.9 ‰, with mostly negligible N isotope fractionations between NO and NO₂ due to the high $NO_2/NO_x$ ratios. After correction of nitrogen isotopic fractionations, we use a Bayesian isotope mixing model to estimate the relative contributions of the dominant $NO_x$ emissions sources. The results indicate the predominance of traffic $NO_x$ emissions in this area at $(57 \pm 8)$ %, before natural gas combustion and soil emission.

Despite the limited nature of our measurements dataset, our results shed light on the sensitivity of NO₂ isotopic signature to the atmospheric chemical regimes and emissions of the local environment. This isotopic approach can be applied to various environments in order to probe further the oxidative chemistry and help to constrain the $NO_x$ fate in a more quantitative way. In the future, the interpretation of the isotopic data should be extended with the use of a photochemical box model including isotopic anomaly transfers and local emissions in order to solve persistent issues of atmospheric oxidation mechanisms. Moreover, samplings and multi-isotopic analysis of atmospheric nitrate performed in parallel to those of NO₂ would certainly be of interest for the study of the full reactive nitrogen cycle.

## Appendix A: Isotopic standards and calibration

This method of analysis induces isotope fractionations during $NO_2^-/N_2O$ conversion and ionization in the spectrometer, as well as isotope exchanges between $NO_2^-$ and its medium. Indeed, while isotope exchanges between nitrite and its matrix are minimized due to the basic pH, the chemistry required to convert nitrite to N₂O involves a step in an acidic medium that promotes an exchange of oxygen isotopes (Casciotti et al., 2007). In order to eliminate the effects of these isotope splits, the system is calibrated using standards of known isotopic composition, which are subjected to the same treatment as the samples. This is called the identical treatment principle (Brand, 1996). By subjecting compounds of known isotopic composition to the same treatment as the samples, the isotope fractionation induced by the analytical protocol can be estimated and the samples values can be corrected. Standards are first dissolved in a basic aqueous medium (pH = 12) and then, from this stock solution, five series of each standard are prepared in several concentration ranges, namely, 40 nmol, 80 nmol, 100 nmol, 120 nmol and 150 nmol, in order to estimate the effects of the concentration of a material on its isotopic measurement. The matrix used for their preparation is the same as that of the samples, i.e. a mixture of KOH 2 M/guaiacol in deionised water. Correction factors are obtained by linear regression between the raw and the expected values of $\delta^{15}N$, $\delta^{18}O$ and $\delta^{17}O$ of the standards. Three international references of known $\delta^{15}N$ and $\delta^{18}O$ values are used for this work. These are nitrite salts, named RSIL-N7373, RSIL-N10219 and RSIL-N23 with respective $\delta^{15}N/\delta^{18}O$ values of −79.6/4.2 ‰, 2.8/88.5 ‰, and 3.7/11.4 ‰. Although the three standards cover a wide range of isotopic composition in $\delta^{15}N$ and $\delta^{18}O$, they do not have an isotopic anomaly in $^{17}O$. As we are not aware of any available international reference nitrite standards with a known $^{17}O$ anomaly, we are currently in the process of manufacturing our own standards. As this step is still under development, and in order to be able to assess the accuracy of our $^{17}O$ measurements of atmospheric NO₂ samples, we have estimated the isotope fractionation that $^{17}O$ undergoes during the analysis. RSIL-N7373 and RSIL-N23 standards have a $\Delta^{17}O = 0$ ‰ so we estimate their $^{17}O$ composition such that

$\delta^{17}O = 0.52 \times \delta^{18}O$. For standard RSIL-N10219, we measure a negative $\Delta^{17}O$ around −7 ‰. We therefore apply the mass

independent relation such that $\delta^{17}O_{std}(RSIL\text{-}N10219) = \Delta^{17}O_{raw}(RSIL\text{-}N10219) + 0.5 \times \delta^{18}O_{std}(RSIL\text{-}N10219)$.

The isotopic exchange of $^{18}O$ is estimated at 11 % for standards at 100 nmol (Fig. A1) which is in line with Kobayashi et al.

(2020) who have estimated the degree of O isotope exchange in the azide method between $H_2O$ and $NO_2^-$ to $(10.8 \pm 0.3)$ %.

The $^{15}N$ calibration curve allows us to ensure a good fractionation rate during the analysis. Indeed, given the 1:1 association

of the nitrogen atoms of nitrite and azide, the theoretical value of the calibration slope must be 0.5. The slight deviation from

our measured value can be attributed to a blank effect, estimated here at 2 % of the size of the standards (6 % for those at 40

nmol).

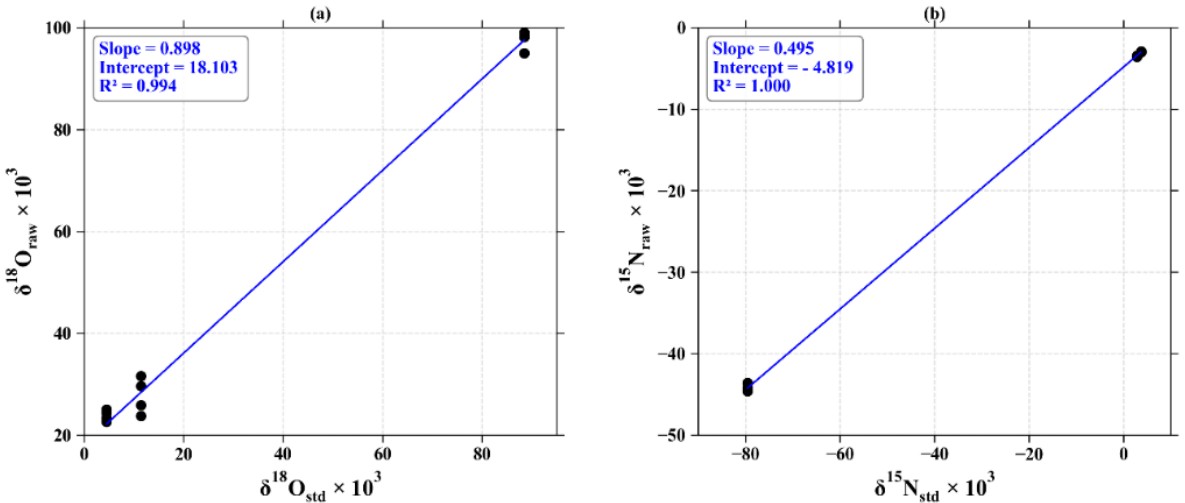

**Figure A1.** Calibration of **(a)** $^{18}O$ and **(b)** $^{15}N$ with nitrite standards at 100 nmol measured by the chemical azide method. The measured $\delta^{18}O$ ($\delta^{18}O_{raw}$) and $\delta^{15}N$ ($\delta^{15}N_{raw}$) values of $NO_2^-$ standards are plotted against their certified reference $\delta^{18}O$ ($\delta^{18}O_{std}$) and $\delta^{15}N$ ($\delta^{15}N_{std}$) values.

### Appendix B: Samples isotopic stability

Oxygen isotopes in nitrites are very labile (Böhlke et al., 2007) but the basic pH of the eluent limits isotopic exchanges. To

ensure isotopic integrity from denuders extraction to analysis by IRMS, we followed Walters et al. (2018) procedure to quantify

isotopic exchanges that might occur with the eluted matrix during storage. Thus, three solutions containing each 500 nmol of

a $KNO_2$ salt (RSIL-N7373, RSIL-N10219 and RSIL-N23) were prepared in the eluted matrix and kept frozen. We monitored

the nitrite standards isotopic composition prepared in the eluted guaiacol matrix during 22 days. 100 nmol were collected from

the individual solutions, analysed and refrozen until the next analysis. The temporal evolution of the $\delta^{17}O$, $\delta^{18}O$ and $\Delta^{17}O$

differences between our measurements of RSIL standards (prepared in the KOH/guaiacol eluted matrix) and their certified

reference values is plotted in Fig. B1. It represents the temporal drift of the isotopic signal with respect to reference values. If

the deviation is constant, it means that the isotopic signal is not degraded with time and its standard deviation is considered as the uncertainty in our $\delta^{17}O(NO_2)$ and $\delta^{18}O(NO_2)$ measurements. As shown in Fig. B1, deviation of the three standards was stable over the 22-days experiment with an overall mean of $(1.1 \pm 0.8)$ ‰, $(2.3 \pm 1.8)$ ‰, and $(-0.1 \pm 0.3)$ ‰ for $\delta^{17}O$, $\delta^{18}O$

and $\Delta^{17}O$, respectively. Note that RSIL-N10219 shows higher $\delta^{17}O$ and $\delta^{18}O$ residuals than the two other standards. The reason for this difference of behaviour is still not fully understood. As residuals remain steady over several weeks, we consider this method suitable for the oxygen analysis of $NO_2$ and the uncertainties applied to our isotopic measurements are reported as the propagation error of the mean measurement uncertainty and the mean uncertainty resulting from $NO_2^-$ storage. In our study, average uncertainties on $\delta^{17}O$, $\delta^{18}O$, and $\Delta^{17}O$ are estimated to be $\pm 1.1$, $\pm 2.5$ and, $\pm1.7$ ‰, respectively (1σ uncertainties).


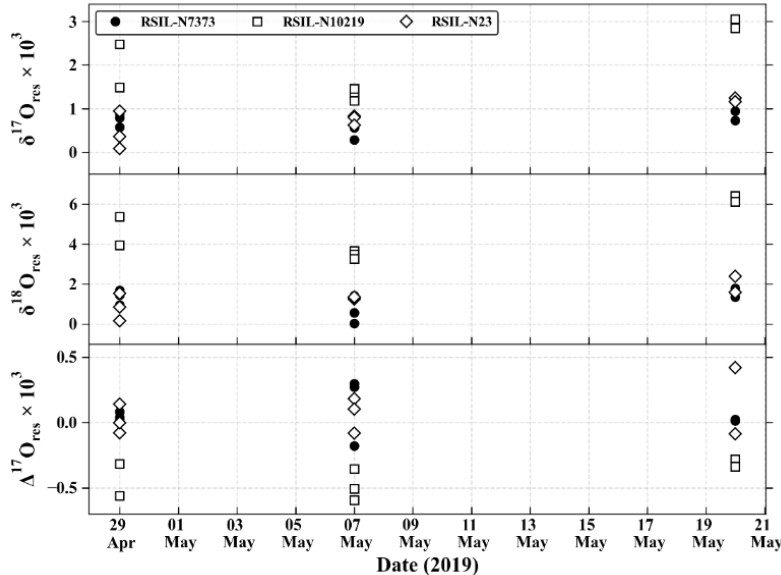

**Figure B1.** Temporal evolution of $\delta^{17}O$, $\delta^{18}O$ and, $\Delta^{17}O$ differences between our measurements of RSIL standards (prepared in the KOH/guaiacol eluted matrix) and their certified reference values. Error bars derived from measurement uncertainties are approximately equivalent to the size of the markers.


**Appendix C: Deriving the N isotopic fractionation from isotopic exchange and the extended Leighton cycle**

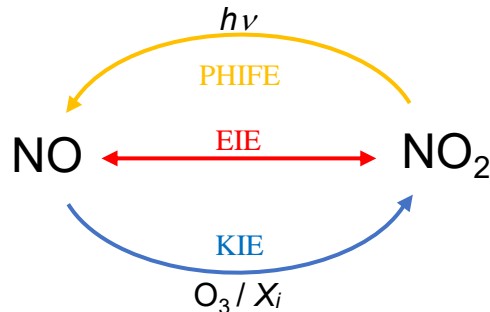

**Figure C1.** Sketch of the nitrogen fractionation processes between NO and $NO_2$. PHIFE for Photochemical Isotope Fractionation Effect, KIE for Kinetic Isotope Effect and EIE for Equilibrium Isotope Effect.


We follow the same approach as Li et al. (2020) but take into account all the oxidation pathways of NO into $NO_2$, not only via $O_3$. The reactions considered in deriving the combined isotopic fractionation are the following:

$$^{15}NO_2 + {}^{14}NO \rightarrow {}^{14}NO_2 + {}^{15}NO \qquad\qquad k_{NO+NO_2} \qquad\qquad (CR1)$$

$$^{14}NO_2 + {}^{15}NO \rightarrow {}^{15}NO_2 + {}^{14}NO \qquad\qquad k_{NO+NO_2} \times \alpha_{EIE} \qquad\qquad (CR2)$$

$$^{14}NO_2 \rightarrow {}^{14}NO + O \qquad\qquad J_{NO_2} \qquad\qquad (CR3)$$

$$^{15}NO_2 \rightarrow {}^{15}NO + O \qquad\qquad J_{NO_2} \times \alpha_{PHIFE} \qquad\qquad (CR4)$$

$$^{14}NO + O_3 \rightarrow {}^{14}NO_2 + O_2 \qquad\qquad k_{NO+O_3} \qquad\qquad (CR5)$$

$$^{15}NO + O_3 \rightarrow {}^{15}NO_2 + O_2 \qquad\qquad k_{NO+O_3} \times \alpha_{KIE(NO+O_3)} \qquad\qquad (CR6)$$

$$^{14}NO + X_i \rightarrow {}^{14}NO_2 + O_2 \qquad\qquad k_{NO+X_i} \qquad\qquad (CR7)$$

$$^{15}NO + X_i \rightarrow {}^{15}NO_2 + O_2 \qquad\qquad k_{NO+X_i} \times \alpha_{KIE(NO+X_i)} \qquad\qquad (CR8)$$

with $X_i$ = $RO_2$, BrO, ClO …

**Daytime N fractionation**

During the day, $NO_2$ photolysis is the overwhelmingly dominant $NO_2$ sink and NO oxidation is the main $NO_2$ source. The assumption of steady-state on $^{15}NO_2$ for the extended Leighton cycle leads to


$$k_{NO+NO_2}[^{15}NO_2][^{14}NO] + J_{NO_2}\,\alpha_{PHIFE}\,[^{15}NO_2] =$$
$$\Sigma(k_{NO+X_i}\,\alpha_{KIE(NO+X_i)}[^{15}NO][X_i]) + k_{NO+NO_2}\,\alpha_{EIE}\,[^{14}NO_2][^{15}NO] \tag{C1}$$

where $k_{NO+NO_2}$ is the rate constant for the nitrogen isotopic exchange between NO and $NO_2$, $J_{NO_2}$ the $NO_2$ photolysis rate with $\alpha_{PHIFE}$ its isotopic fractionation factor, $\Sigma\,k_{NO+X_i}[X_i]$ the sum of all the NO oxidation pathways to $NO_2$, $X_i$ the NO oxidant (i.e. $O_3$, $RO_2$, BrO, ClO…), and $k_{NO+X_i}$ the rate constant for the reaction of $NO + X_i$ with $\alpha_{KIE(NO+X_i)}$ its isotopic fractionation

factor. C1 can be rearranged to give

$$\frac{[^{15}NO_2]}{[^{15}NO]} = \frac{\Sigma(k_{NO+X_i}\,\alpha_{KIE(NO+X_i)}[X_i]) + k_{NO+NO_2}\,\alpha_{EIE}\,[^{14}NO_2]}{k_{NO+NO_2}[^{14}NO] + J_{NO_2}\,\alpha_{PHIFE}} \tag{C2}$$

Meanwhile, $^{14}NO_2$ in steady-state leads to

$$\frac{[^{14}NO_2]}{[^{14}NO]} = \frac{\Sigma\,k_{NO+X_i}[X_i]}{J_{NO_2}} \tag{C3}$$

We define $A^*{}_{day}$ as the ratio of the $^{14}NO_2$ lifetime with respect to isotopic exchange with $^{14}NO$ ($\tau_{exchange-NO_2}$) over the daytime

$^{14}NO_2$ chemical lifetime ($\tau_{chem-NO_2}$) (Li et al., 2020):

$$A^*{}_{day} = \frac{\tau_{exchange-NO_2}}{\tau_{chem-NO_2}} = \frac{J_{NO_2}}{k_{NO+NO_2}[^{14}NO]} \tag{C4}$$

Using C3, C4 becomes

$$A^*{}_{day} = \frac{\Sigma\,k_{NO+X_i}[X_i]}{k_{NO+NO_2}[^{14}NO_2]} \tag{C5}$$

We also define $T_{NO+X_i}$ the relative importance of the oxidation pathway of NO into $NO_2$ via the oxidant $X_i$:


$$T_{NO+X_i} = \frac{k_{NO+X_i}[X_i]}{\Sigma\,k_{NO+X_i}[X_i]} \tag{C6}$$

with necessarily $\Sigma\,T_{NO+X_i} = 1$.

Using the definitions C5 and C6, C2 becomes

$$\frac{[^{15}\text{NO}_2]}{[^{15}\text{NO}]} = \frac{A^*{}_{\text{day}}\, k_{\text{NO}+\text{NO}_2}[^{14}\text{NO}_2]\alpha_{\text{KIE}} + k_{\text{NO}+\text{NO}_2}\,\alpha_{\text{EIE}}[^{14}\text{NO}_2]}{k_{\text{NO}+\text{NO}_2}[^{14}\text{NO}] + A^*{}_{\text{day}}\, k_{\text{NO}+\text{NO}_2}[^{14}\text{NO}]\alpha_{\text{PHIFE}}} \tag{C7}$$

with $\alpha_{\text{KIE}} = \sum T_{\text{NO}+\text{X}_i} \times \alpha_{\text{KIE(NO}+\text{X}_i)}$

Using $R(^{15}\text{N}/^{14}\text{N}, \text{NO}) = R_{\text{NO}}(^{15}\text{N})/R_{\text{NO}}(^{14}\text{N}) = {}^{15}R_{\text{NO}}$   (with   $\delta^{15}\text{N(NO)} = {}^{15}R_{\text{NO}}/^{15}R_{\text{standard}} - 1$   and   $R(^{15}\text{N}/$
$^{14}\text{N}, \text{NO}_2) = R_{\text{NO}_2}(^{15}\text{N})/R_{\text{NO}_2}(^{14}\text{N}) = {}^{15}R_{\text{NO}_2}$ (with $\delta^{15}\text{N(NO}_2) = {}^{15}R_{\text{NO}_2}/^{15}R_{\text{standard}} - 1$), C7 becomes

$$\frac{^{15}R_{\text{NO}_2}}{^{15}R_{\text{NO}}} = \frac{[^{15}\text{NO}_2][^{14}\text{NO}]}{[^{15}\text{NO}][^{14}\text{NO}_2]} = \frac{A^*{}_{\text{day}}\,\alpha_{\text{KIE}} + \alpha_{\text{EIE}}}{1 + A^*\alpha_{\text{PHIFE}}} \tag{C8}$$

$$\frac{^{15}R_{\text{NO}}}{^{15}R_{\text{NO}_2}} - 1 = \frac{A^*{}_{\text{day}}(\alpha_{\text{PHIFE}} - \alpha_{\text{KIE}}) - (\alpha_{\text{EIE}} - 1)}{A^*{}_{\text{day}}\,\alpha_{\text{KIE}} + \alpha_{\text{EIE}}} \tag{C9}$$

As a result, the daytime isotopic shift of NO$_2$ relative to NO, defined as $\Delta(\text{NO}_2 - \text{NO}) = \delta^{15}\text{N(NO}_2) - \delta^{15}\text{N(NO)}$, is given by

$$\Delta_{\text{day}}(\text{NO}_2 - \text{NO}) = \frac{A^*{}_{\text{day}}\,(\alpha_{\text{KIE}} - \alpha_{\text{PHIFE}}) + (\alpha_{\text{EIE}} - 1)}{A^*{}_{\text{day}}\,\alpha_{\text{KIE}} + \alpha_{\text{EIE}}}(1 + \delta^{15}\text{N(NO}_2)) \tag{C10}$$

Using the isotopic balance $\delta^{15}\text{N(NO}_x) = f_{\text{NO}_2}\delta^{15}\text{N(NO}_2) + (1 - f_{\text{NO}_2})\delta^{15}\text{N(NO)}$ with $f_{\text{NO}_2} = [\text{NO}_2]/[\text{NO}_x]$ (Li et al. (2020)), the isotopic shift of NO$_2$ relative to NO$_x$, defined as $\Delta(\text{NO}_2 - \text{NO}_x) = \delta^{15}\text{N(NO}_2) - \delta^{15}\text{N(NO}_x)$, can be expressed by:

$$\Delta_{\text{day}}(\text{NO}_2 - \text{NO}_x) = \frac{A^*{}_{\text{day}}(\alpha_{\text{KIE}} - \alpha_{\text{PHIFE}}) + (\alpha_{\text{EIE}} - 1)}{A^*{}_{\text{day}}\,\alpha_{\text{KIE}} + \alpha_{\text{EIE}}}(1 + \delta^{15}\text{N(NO}_2))(1 - f_{\text{NO}_2}) \tag{C11}$$

Since fractionation factors are close to unity and $1 + \delta^{15}\text{N(NO}_2) \approx 1$, C11 can be further simplified by keeping only the
dominant terms (Li et al. 2020):

$$\Delta_{\text{day}}(\text{NO}_2 - \text{NO}_x) \approx \frac{\alpha_{\text{LCIE}}{}^*\, A^*{}_{\text{day}} + (\alpha_{\text{EIE}} - 1)}{A^* + 1}(1 - f_{\text{NO}_2}) \tag{C12}$$
with $\alpha_{\text{LCIE}}{}^* = \alpha_{\text{KIE}} - \alpha_{\text{PHIFE}}$

Considering the localisation of our sampling site (urban mid-latitude area), only NO + RO$_2$ and NO + O$_3$ are thought to be significant as NO$_2$ formation pathways and hence $\alpha_{\text{LCIE}}$* becomes

$$\alpha_{\text{LCIE}}{}^* = T_{\text{NO}+\text{O}_3} \times \alpha_{\text{KIE(NO}+\text{O}_3)} + T_{\text{NO}+\text{RO}_2} \times \alpha_{\text{KIE(NO}+\text{RO}_2)} - \alpha_{\text{PHIFE}} \tag{C13}$$

and C5 becomes

$$A^*{}_{\text{day}} = \frac{k_{\text{NO}+\text{O}_3}[\text{O}_3] + k_{\text{NO}+\text{RO}_2}[\text{RO}_2]}{k_{\text{NO}+\text{NO}_2}[^{14}\text{NO}_2]} \tag{C14}$$

Eq.4 and Eq.5 from section 4.1.2 can be combined to give

$$k_{NO+O_3}[O_3] + k_{NO+RO_2}[RO_2] = \frac{\Delta^{17}O_{NO+O_3}(NO_2)}{\Delta^{17}O_{day}(NO_2)} k_{NO+O_3}[O_3] \tag{C15}$$

Using C15, C14 becomes

$$A^*_{day} = \frac{k_{NO+O_3}[O_3]}{k_{NO+NO_2}[^{14}NO_2]} \left( \frac{\Delta^{17}O_{NO+O_3}(NO_2)}{\Delta^{17}O_{day}(NO_2)} \right) \tag{C16}$$

We consider several particular cases. The first case is when $\alpha_{KIE(NO+O_3)} \approx \alpha_{KIE(NO+RO_2)}$. Previous studies found that the NO + $O_3$ reaction falls within the family of "normal kinetic isotope fractionation" with the $NO_2$ produced being depleted in $^{15}$N (Walters and Michalski, 2016) compared to residual reactant NO. To our knowledge, no such experiment has been carried out

for the NO + $RO_2$ reaction. Nonetheless, considering the very close, and both very low, activation energies for the reaction NO + $O_3$ and NO + $RO_2$, it is quite likely that the fractionation factors of these two reactions are similar. It follows that we obtain the same expression for $\alpha_{LCIE}^*$ as for the $\alpha_{LCIE}$ given in Li et al. (2020):

$$\alpha_{LCIE}^* = \alpha_{KIE(NO+O_3)} - \alpha_{PHIFE} \tag{C17}$$

And C12 becomes

$$\Delta_{day}(NO_2 - NO_x) = \frac{\alpha_{LCIE}^* A^*_{day} + (\alpha_{EIE} - 1)}{A^*_{day} + 1} (1 - f_{NO_2}) \tag{C18}$$

C18 with $A^*_{day}$ given by C16 is the expression that we use to analyse our daytime nitrogen isotopic measurements. Another particular case considered by Li et al. (2020) is $k_{NO+O_3}[O_3] \gg k_{NO+RO_2}[RO_2]$; in that case, $\alpha_{LCIE}^*$ is still given by C12 but $A^*$ is simplified:

$$A^*_{day} = \frac{k_{NO+O_3}[O_3]}{k_{NO+NO_2}[NO_2]} \tag{C19}$$

C16 with $A^*_{day}$ given by C19 and $\alpha_{LCIE}^*$ given by C17 is the same expression as Eq.8 in Li et al. (2020).

**Nighttime N fractionation**

An expression similar to C1 can be derived for nighttime conditions, when $NO_2$ photolysis is null and hence there is no recycling between NO and $NO_2$. In addition, the conversion of NO into $NO_2$ occurs only via reaction with $O_3$ because the

concentrations of other NO oxidants are usually negligible at night. The main source of $NO_x$ at night is the NO emissions. The

assumption of steady-state on short time scales can only hold for $^{14}$NO and $^{15}$NO, not NO$_2$, leading to an equation equivalent to C1:

$$k_{NO+NO_2}[^{15}NO_2][^{14}NO] + E(^{15}NO)$$

$$= k_{NO+O_3}\alpha_{KIE}[^{15}NO][O_3] + k_{NO+NO_2}\alpha_{EIE}[^{14}NO_2][^{15}NO] \tag{C20}$$

with $E(^{15}NO)$ being the $^{15}$NO emission flux and $\alpha_{KIE}$ is the fractionation factor of NO + O$_3$. C20 can be rearranged to give

$$\frac{[^{15}NO_2]}{[^{15}NO]} = \frac{k_{NO+O_3}\alpha_{KIE}[O_3] + k_{NO+NO_2}\alpha_{EIE}[^{14}NO_2]}{k_{NO+NO_2}[^{14}NO] + E(^{15}NO)/[^{15}NO_2]} \tag{C21}$$

Meanwhile, the steady-state on $^{14}$NO gives (nitrogen isotopic exchanges are neglected as NO emissions are largely dominated by $^{14}$NO):

$$E(^{14}NO) = k_{NO+O_3}[^{14}NO][O_3] \tag{C22}$$

with $E(^{14}NO)$ being the $^{14}$NO emission flux. For nighttime, we define $A^*_{night}$ as the ratio of the $^{14}$NO lifetime with respect to isotopic exchange with $^{14}$NO$_2$ ($\tau_{exchange-NO}$) over the nighttime $^{14}$NO chemical lifetime ($\tau_{chem-NO}$):

$$A^*_{night} = \frac{\tau_{exchange-NO}}{\tau_{chem-NO}} = \frac{k_{NO+O_3}[O_3]}{k_{NO+NO_2}[^{14}NO_2]} \tag{C23}$$

Using C22, C23 gives

$$A^*_{night} = \frac{E(^{14}NO)}{k_{NO+NO_2}[^{14}NO][^{14}NO_2]} \tag{C24}$$

NO$_x$ is overwhelmingly emitted in the form of NO. The isotopic signature of NO emissions can be characterised with $^{15}R_{NO_{emis}}$ and $\delta^{15}N(NO_{emis}) = {}^{15}R_{NO_{emis}}/{}^{15}R_{standard} - 1$.

Using the definition of $^{15}R_{NO_{emis}}$, C23 and C24, C21 becomes

$$\frac{[^{15}NO_2]}{[^{15}NO]} = \frac{A^*_{night} k_{NO+NO_2}[^{14}NO_2]\,\alpha_{KIE} + k_{NO+NO_2}\alpha_{EIE}[^{14}NO_2]}{k_{NO+NO_2}[^{14}NO] + (^{15}R_{NO_{emis}}E(^{14}NO))/[^{15}NO_2]} \tag{C25}$$

And then, using C24, C25 becomes

$$\frac{^{15}R_{NO_2}}{^{15}R_{NO}} = \frac{[^{15}NO_2][^{14}NO]}{[^{15}NO][^{14}NO_2]} = \frac{A^*_{night}\alpha_{KIE} + \alpha_{EIE}}{1 + A^*_{night}(^{15}R_{NO_{emis}}/^{15}R_{NO_2})} \tag{C26}$$

$$\frac{^{15}R_{NO}}{^{15}R_{NO_2}} - 1 = \frac{A^*_{night}\left((^{15}R_{NO_{emis}}/^{15}R_{NO_2}) - \alpha_{KIE}\right) - (\alpha_{EIE} - 1)}{A^*_{night}\alpha_{KIE} + \alpha_{EIE}} \tag{C27}$$

Following the approach used in the derivation of daytime isotopic shift, the nighttime isotopic shift of $NO_2$ relative to NO is given by:

$$\Delta_{night}(NO_2 - NO) = \frac{A^*_{night}\left(\alpha_{KIE} - (^{15}R_{NO_{emis}}/^{15}R_{NO_2})\right) + (\alpha_{EIE} - 1)}{A^*_{night}\alpha_{KIE} + \alpha_{EIE}}(1 + \delta^{15}N(NO_2)) \tag{C28}$$

Using the isotopic balance $\delta^{15}N(NO_x) = f_{NO_2}\delta^{15}N(NO_2) + (1 - f_{NO_2})\delta^{15}N(NO)$, the nighttime isotopic shift of $NO_2$ relative to $NO_x$, can be expressed by:

$$\Delta_{night}(NO_2 - NO_x) = \frac{A^*_{night}\left(\alpha_{KIE} - \left(\frac{1 + \delta^{15}N(NO_{emis})}{1 + \delta^{15}N(NO_2)}\right)\right) + (\alpha_{EIE} - 1)}{A^*_{night}\alpha_{KIE} + \alpha_{EIE}}(1 + \delta^{15}N(NO_2))(1 - f_{NO_2}) \tag{C29}$$

where $\delta^{15}N(NO_{emis})$ is the nitrogen isotopic composition of NO emissions.

Keeping the dominant terms, C29 can be further simplified following the daytime derivation:

$$\Delta_{night}(NO_2 - NO_x) \approx \frac{A^*_{night}\left(\alpha_{KIE} - \left(\frac{1 + \delta^{15}N(NO_{emis})}{1 + \delta^{15}N(NO_2)}\right)\right) + (\alpha_{EIE} - 1)}{A^*_{night} + 1}(1 - f_{NO_2}) \tag{C30}$$

We consider two particular cases. When $A^*_{night} \ll 1$ ($k_{NO+NO_2}[^{14}NO_2] \gg k_{NO+O_3}[O_3]$) i.e isotopic exchange much faster than NO oxidation, C28 becomes

$$\Delta_{night}(NO_2 - NO) = \frac{(\alpha_{EIE} - 1)}{\alpha_{EIE}}(1 + \delta^{15}N(NO_2)) \tag{C31}$$

Keeping the dominant terms, C31 can be simplified

$$\Delta_{night}(NO_2 - NO) \approx (\alpha_{EIE} - 1) \tag{C32}$$

As expected, the nighttime isotopic shift of $NO_2$ relative to NO depends only on the isotopic exchange fractionation in that case. In the same way, C29 becomes

$$\Delta_{night}(NO_2 - NO_x) = \frac{(\alpha_{EIE} - 1)}{\alpha_{EIE}}(1 + \delta^{15}N(NO_2))(1 - f_{NO_2}) \tag{C33}$$

Keeping the dominant terms, C33 can be simplified

$$\Delta_{night}(NO_2 - NO_x) \approx \frac{(\alpha_{EIE} - 1)}{\alpha_{EIE}}(1 - f_{NO_2}) \tag{C34}$$

When $A^*_{night} \gg 1$ ($k_{NO+NO_2}[^{14}NO_2] \ll k_{NO+O_3}[O_3]$) i.e NO oxidation much faster than isotopic exchange, C28 yields

$$\Delta_{night}(NO_2 - NO) = 1 + \delta^{15}N(NO_2) - \left(\frac{1 + \delta^{15}N(NO_{emis})}{\alpha_{KIE}}\right) \tag{C35}$$

leading to

$$1 + \delta^{15}N(NO) = \left(\frac{1 + \delta^{15}N(NO_{emis})}{\alpha_{KIE}}\right) \tag{C36}$$

$$^{15}R_{NO} = \frac{^{15}R_{NO_{emis}}}{\alpha_{KIE}} \tag{C37}$$

As expected, the nighttime isotopic shift of NO relative to NO emissions depends only on the isotopic fractionation factor of the NO + O$_3$ reaction in that case. It is possible to estimate the nighttime isotopic shift of NO$_2$ relative to NO or NO$_x$ on long timescales by assuming crudely that $^{14}NO_2$ is in steady-state:

$$k_{loss-NO_2}[^{14}NO_2] = k_{NO+O_3}[^{14}NO][O_3] \tag{C38}$$

with $k_{loss-NO_2}$ representing the equivalent of a first-order rate constant. If the $^{14}NO_2$ loss is a second-order reaction such as NO$_2$ + O$_3$ loss, $k_{loss-NO_2} = k_{NO_2+O_3}[O_3]$. In the same way, assuming that the NO$_2$ oxidation into nitrate via O$_3$ is not fractionating, $^{15}NO_2$ in steady-state gives:

$$k_{loss-NO_2}[^{15}NO_2] = k_{NO+O_3}\alpha_{KIE}[^{15}NO][O_3] \tag{C39}$$

Using C38, C39 becomes

$$^{15}R_{NO_2} = \alpha_{KIE}\ ^{15}R_{NO} \tag{C40}$$

Using C37, C40 becomes

$$^{15}R_{NO_2} = {}^{15}R_{NO,emis} \tag{C41}$$

Or

$$\delta^{15}N(NO_2) = \delta^{15}N(NO_{emis}) \hspace{3cm} (C42)$$

Under those conditions (negligible isotopic exchange), a measurement of $\delta^{15}N(NO_2)$ is a measurement of $\delta^{15}N(NO_{emis})$, preferably towards the end of the night in order for $^{14}NO_2$ to move towards steady-state.

In the text, $^{14}NO_2$ and $^{14}NO$ are referred as $NO_2$ and $NO$ for convenience.

## Appendix D: kinetic data used


| reaction number | Reactions | Rate constants /cm³ molecule⁻¹ s⁻¹ | References |
|---|---|---|---|
| R3 | $NO + O_3 \rightarrow NO_2 + O_2$ | $k_{NO+O_3} = 1.4 \times 10^{-12} \exp(-1310/T)$ | Atkinson et al. (2004) |
| R4 | $NO + RO_2 \rightarrow NO_2 + RO$ | $k_{NO+RO_2} = 2.3 \times 10^{-12} \exp(360/T)$ | Atkinson et al. (2006) |
| R6 | $NO_2 + O_3 \overset{M}{\rightarrow} NO_3 + O_2$ | $k_{NO_2+O_3} = 1.4 \times 10^{-13} \exp(-2470/T)$ | Atkinson et al. (2004) |
| CR1 | $^{15}NO_2 + {}^{14}NO \rightarrow {}^{14}NO_2\ {}^{15}NO$ | $k_{NO+NO_2} = 8.14 \times 10^{-14}$ | Sharma et al. (1970) |

**Table D1.** Rate constants used for calculations

*Author contribution.* Sampling and analysis protocol were developed by SA under the supervision of JS. NC and AB contributed with technical and knowledge support to SA for isotopic

mass spectrometry and more general atmospheric measurements. SB and JS, supervisors of SA PhD Thesis, helped SA in interpreting the results and writing the manuscript.

*Competing interests.* The authors have no conflict of interests to report.

*Acknowledgements.* This work benefited from the IGE infrastructures and laboratory platforms. This is the publication number 2 of the PANDA platform on which isotope analyses were performed, partially supported by the ANR project ANR-15-IDEX-02 and Labex OSUG@2020, Investissements d'avenir – ANR10 LABX56. The authors acknowledge the support of the ALPACA program (Alaskan Layered Pollution and Arctic Chemical Analysis) funded by two French research organisms: the French polar institute (IPEV, Institut polaire français Paul-Emile Victor) and INSU-CNRS (National Institute of Sciences of

the Universe) via its national LEFE program (Les Enveloppes Fluides et l'Environnement). Finally, the authors thank E. Gauthier, S. Darfeuil and P. Akers for help with laboratory work and more general scientific discussions.

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
