# Peer review of "Measurement report: Nitrogen isotopes ( $\delta^{15}\text{N}$ ) and first quantification of oxygen isotope anomalies ( $\Delta^{17}\text{O}$ , $\delta^{18}\text{O}$ ) in atmospheric nitrogen dioxide"

_Atmospheric Chemistry and Physics, 2020_

## Referee Comment (RC1) · Anonymous Referee #1 · 11 Jan 2021

Summary: The author's report on the multiple oxygen and nitrogen stable isotopes of ambient NO2 from an urban location in France. The nitrogen and oxygen stable isotope signatures of NOx and its oxidation products are a potentially valuable tool to further understand the emission and chemistry of this important reactive nitrogen family; however, traditionally, isotopic measurements have been made almost exclusively for atmospheric nitrate and nitric acid (i.e., secondary products from NOx emission). While this dataset isn't the first  $\delta$ 15N or  $\delta$ 18O measurement of ambient NO2, it builds off a previous study using a similar collection technique and reports the first  $\Delta$ 17O that

seems unaffected from collection artifacts. This dataset will certainty help further our understanding of the isotope dynamics of NO2 and its propagation into atmospheric nitrate. The biggest drawback to this study was the limited nature of the dataset that only included measurements for one day at one site. The limited dataset may be appropriate for a measurement report; however, I think interpretations of the dataset should be treated cautiously because it's not clear how representative one day's worth of measurements would be for our understanding of diurnal NO2 chemistry or emission contributions. Despite this limitation, I found the analyses and interpretation to be robust and I think is worthy of publication with the following minor comments/suggestions:

Specific Comments/Suggestions:

Lines 32-34: In the introduction, I think the authors should point out that there is also motivation to better constrain precursor emission contributions to nitrate deposition; thus, source apportionment is also important (and not just chemistry).

Line 243:  $\delta$ 15N(NO2) range looks to be incorrect; I think it should be -11.8 to -4.9 ‰ (based on Table 1).

Line 280: EPA IsoSource is a very simplistic model that cannot account for source uncertainty. I think the authors should consider applying a more advanced statistical (i.e., Monte-Carlo) mixing model such as SIMR or SIAR that has been commonly used in the  $\delta$ 15N atmospheric community for the past few years. As a measurement report, I think it is important to showcase how advanced statistical modeling and be used to partition NOx emission sources using the described sampling technique.

Lines 284-286: In recent years, there have been several updates to our  $\delta$ 15N(NOx) source emission values including for biogenic emissions (rural and urban; Yu and Elliott, ES&T, 2017; Miller et al., GRL, 2018) and traffic (Miller et al., JGR:Atmos, 2017). Perhaps consider using more up to date  $\delta$ 15N(NOx) values. Additionally, the fuel-combustion signature is for natural gas power plants. Please confirm that is an appropriate fuel-combustion source signature for your study region.

Lines 376-377: Can you further elaborate and include specific details on the "additional more accurate measurements" that are needed to improve the interpretation of NO + RO2 rxn contributions to  $\Delta$ 17O?

General: As mentioned in the summary section, the authors report on isotopic measurements over a limited collection period (1 day during at a single collection location). I think the authors should tone down their NOx source contributions and chemical mechanism conclusions based on their isotopic measurements. For example, NOx emission contributions are going to be highly dependent on meteorology conditions (wind speed and wind direction); thus, measurements from one day are unlikely to capture even the seasonal NOx emission patterns at their sampling location. Therefore, I was surprised the authors used their limited dataset to draw conclusions about urban NOx emissions compared to satellite observations. Instead I think the focus should be on explaining the framework for interpreting  $\delta 15N$  and  $\Delta 17O(+\delta 18O)$  and what could be learned from these measurements without drawing large conclusions due to their limited dataset.

СЗ

---

## Referee Comment (RC2) · Anonymous Referee #2 · 5 Mar 2021

This paper provides first results and their interpretation from an atmospheric NO2 sampling technique that includes isotopic analysis of double nitrogen and triple oxygen isotopes. The technique is deployed in an urban location (Grenoble, France) over a short ~1.5 day period, providing 3-hourly simultaneous isotopic composition measurements for N and O isotopes. Despite the relatively short deployment, the paper serves as an important demonstration of this powerful technique, and the analysis draws conclusions on source influences on NOx in this environment (albeit over this limited period). The paper is well written and structured, with clear figures. My view is that the paper is suitable for publication in ACP, once the following minor comments have been addressed.

Line 80: Could add reference to e.g. Michoud et al., (2014) which is a more recent specific study on HONO and NOy relevant to an urban France location (Paris).

From what I understand, Eq 3 assumes that NO —> NO2 conversion is dominated by O3? Do the authors have information available to clarify this here? i.e. is the observed NO/NO2 ratio close to that given by the Leighton ratio approximation based on NO2 photolysis and NO+O3? Is the discussion on lines 370-373 also relevant here?

Line 267: Is there any kinetic / mechanistic theoretical explanation that supports the suggestion that NO+RO2 fractionation is similar to that of NO+O3? Might be worth mentioning here if so.

Also, can the authors comment on the similarity of the chemical environment in the Li et al., (2020) study to their study location?

Line 292-299 - Comparison with COVID lockdown satellite study seems a bit tenuous (i.e. comparing NO2 column change over a large city to limited time measurements here). Are there other estimates of traffic contributions to urban NOx that could be compared?

Line 325: The reaction NO2 + OH to produce HNO3 is termolecular, involving a third body. i.e. NO2 + OH + M —> HNO3 + M. See e.g. Atkinson et al., (2004).

Line 369: Derivation of [RO2] and discussion. The diurnal behaviour of the derived RO2 seems surprising. What is assumed for k_NO+RO2 in Equation 10? How sensitive is RO2 to the assumed speciation of RO2? i.e. is the value assumed to simply be that for HO2 or is there some weighting for an assumed VOC mixture, and does this matter much? In general it might be useful to provide a Table (in the Appendix?) of values and sources of rate constant values used.

Typographical errors:

Line 57: ang = and

References

Atkinson, R., Baulch, D. L., Cox, R. A., Crowley, J. N., Hampson, R. F., Hynes, R. G., Jenkin, M. E., Rossi, M. J., and Troe, J.: Evaluated kinetic and photochemical data for atmospheric chemistry: Volume I - gas phase reactions of Ox, HOx, NOx and SOx species, Atmos. Chem. Phys., 4, 1461–1738, https://doi.org/10.5194/acp-4-1461-2004, 2004.

Michoud, V., Colomb, A., Borbon, A., Miet, K., Beekmann, M., Camredon, M., Aumont, B., Perrier, S., Zapf, P., Siour, G., Ait-Helal, W., Afif, C., Kukui, A., Furger, M., Dupont, J. C., Haeffelin, M., and Doussin, J. F.: Study of the unknown HONO daytime source at a European suburban site during the MEGAPOLI summer and winter field campaigns, Atmos. Chem. Phys., 14, 2805–2822, https://doi.org/10.5194/acp-14-2805-2014, 2014.

---

## Author Comment (AC1) · 9 May 2021

**Response to the anonymous referee #1 (ACP-2020-1143)**

We thank anonymous referee #1 for his/her positive review of this work and relevant comments. Anonymous referee #1's comments/suggestions are given below in bold followed by our replies.

**1) Lines 32-34: In the introduction, I think the authors should point out that there is also motivation to better constrain precursor emission contributions to nitrate deposition; thus, source apportionment is also important (and not just chemistry).**

Agreed. The sentence has been changed to : "In order to better understand the reactive nitrogen (which includes $NO_x$ and $HNO_3$) chemistry, the related AOC, and the contributions of precursors emissions to nitrate deposition, it is necessary to better constrain $NO_x$ sources and individual chemical processes.". The revised version of the manuscript has been modified accordingly.

**2) Line 243: $\delta^{15}N(NO_2)$ range looks to be incorrect; I think it should be -11.8 to -4.9 ‰ (based on Table 1).**

Thank you for pointing this typing mistake out. The range has been corrected accordingly.

**3) Line 280: EPA IsoSource is a very simplistic model that cannot account for source uncertainty. I think the authors should consider applying a more advanced statistical (i.e., Monte-Carlo) mixing model such as SIMR or SIAR that has been commonly used in the $\delta^{15}N$ atmospheric community for the past few years. As a measurement report, I think it is important to showcase how advanced statistical modelling and be used to partition $NO_x$ emission sources using the described sampling technique.**

Thank you for pointing this out. As recommended, we performed a new estimation of the $NO_x$ sources using the Bayesian isotope mixing model SIAR. Compared to the IsoSource simulation, for which the early morning rush hours sample was dissociated from the rest of the sampling period, we have considered $\delta^{15}N$ measurements as one group for the SIAR simulation. Based on a local $NO_x$ emission inventory and energy balance, we have decided to consider three $NO_x$ sources in our analysis: soil emissions, natural gas combustion and vehicle exhausts. Considering the time of the year (mid-march), we excluded $NO_x$ emissions from biomass burning for home heating.
SIAR simulation results do not change much the overall interpretation of $NO_x$ sources that might influence our site. Like for IsoSource simulation, traffic still being the major contributor in front of natural gas combustion and soil (($57 \pm 8$), ($36 \pm 12$) and ($7 \pm 5$) % respectively). The manuscript has been modified according to the new simulation results.

As you have pointed out, the dataset use for this paper is limited to one site during only one day of sampling. Therefore, a lot of caution has to be exercised when interpreting these measurements. As recommended, we have removed the lines where we compared the IsoSource simulation results to satellite data and focused on the method validity without speculating or generalising any early conclusions.

**4) Lines 284-286: In recent years, there have been several updates to our $\delta^{15}N(NO_x)$ source emission values including for biogenic emissions (rural and urban; Yu and Elliott, ES&T, 2017; Miller et al., GRL, 2018) and traffic (Miller et al., JGR:Atmos, 2017). Perhaps consider using more up to date $\delta^{15}N(NO_x)$ values. Additionally, the fuel-combustion signature is for natural gas power plants. Please confirm that is an appropriate fuel-combustion source signature for your study region.**

Following your comments, the SIAR simulation on estimating the relative contribution of $NO_x$ sources was performed with updated $\delta^{15}N(NO_x)$ source emission values.

As previous studies of vehicles exhausts showed that the variability of $\delta^{15}N$ depends on the fuel type, the reduction emission technology, and the vehicle run time with values ranging from $-21$ ‰ to $-2$ ‰. We use the value of Miller et al. (2017) who have estimated the U.S. vehicle-fleet $NO_x$ isotopic source signature to $(-4.7 \pm 1.7)$ ‰ (integrated on 50-100 km during daytime summer conditions). We think this value can be to some extent representative of our sampling location, as 90 % of the Grenoble vehicle-fleet is composed of diesel-powered engines (85 % for the U.S. vehicle-fleet). According to Grenoble urban area emission inventory, $NO_x$ emissions can be attributed to industries for 26 % and to the residential/tertiary sectors for 20 %. Local energy consumption indicates that industries are powered at 51 % by electricity (mainly produced by nuclear power plants and hydropower dams) and 34 % by natural gas combustion. Additionally, the two main $NO_x$ energy emitters in the residential/tertiary sectors are biomass burning and natural gas combustion. As biomass burning, mainly use for home heating, can be considered negligible at this time of the year, we consider natural gas as the main $NO_x$ emitter for both industry and residential/tertiary sectors. We use the characteristic $\delta^{15}N$ signature of natural gas combustion determined by Walters et al. (2015) $((-16.5 \pm 1.7)$ ‰). In view of the large variability in the isotopic signature of biogenic $NO_x$ emissions reported in the literature (from $-59.8$ to $-19.9$ %), we use a mean value of $(-33.8 \pm 12.2)$ ‰ as reported by Zong et al. (2017). The revised version of the manuscript has been modified accordingly.

**5) Lines 376-377: Can you further elaborate and include specific details on the "additional more accurate measurements" that are needed to improve the interpretation of NO +RO$_2$ rxn contributions to $\Delta^{17}O$**

To study the nitrogen chemistry and test the isotopic approach, the monitoring of atmospheric species as peroxy radicals, $NO_3$ radical or $N_2O_5$, require state of the art instruments and an important technical development. We believe the method presented in this paper can bring a reliable complementary tool for studying the reactive nitrogen chemistry along with studies using the

classic "kinetic method" and which is easier to implement on the field. Nonetheless, to carry out reliable kinetic calculations from these isotopic measurements, we need to monitor precisely $NO_x$ concentrations i.e. with a precision higher our very close to 1 ppb. As it was not the case during our campaign, an important recommendation for further investigations is to conduct isotopic measurements with precise atmospheric chemistry monitoring, at least for $NO_x$ and $O_3$ concentrations. Additionally, the use of a chemical box-model is also recommended because it will allow to account for non-equilibrium effects in isotopic transfers and thus strengthen the interpretation of isotopic measurements in investigations of the nitrogen cycle in urban atmospheres. The revised version of the manuscript has been improved following your comment.

In the marks reviewed version, you will find in red the main modifications from the first version following your comments and the ones of reviewer #2. Additionally, we exchanged the order of the sub-sections of section 4 (Discussion of the multi-isotopic composition of atmospheric $NO_2$). This is because we have developed more general expressions for daytime and nighttime nitrogen isotopic fractionation. The interesting point of the daytime expression is that, despite the absence of $RO_2$ measurements, the $NO + RO_2$ pathway can be accounted for in the estimation of nitrogen isotopic fractionation using $\Delta^{17}O$ measurements. Thus, the description of $\Delta^{17}O$ values must be presented before the section concerning $\delta^{15}N$. We also added two appendixes (C and D). The first appendix presents the derivations of the more general nitrogen isotopic fractionation equations following Li et al., 2020, and the second appendix provides a table of the kinetic constants we use for our calculations.

**References**

Miller, D. J., Wojtal, P. K., Clark, S. C., and Hastings, M. G.: Vehicle $NO_x$ emission plume isotopic signatures: Spatial variability across the eastern United States, 122, 4698–4717, https://doi.org/10.1002/2016JD025877, 2017.

Walters, W. W., Tharp, B. D., Fang, H., Kozak, B. J., and Michalski, G.: Nitrogen isotope composition of thermally produced $NO_x$ from various fossil-fuel combustion sources, 49, 11363–11371, https://doi.org/10.1021/acs.est.5b02769, 2015.

Zong, Z., Wang, X., Tian, C., Chen, Y., Fang, Y., Zhang, F., Li, C., Sun, J., Li, J., and Zhang, G.: First Assessment of $NO_x$ Sources at a Regional Background Site in North China Using Isotopic Analysis Linked with Modeling, Environ. Sci. Technol., 51, 5923–5931, https://doi.org/10.1021/acs.est.6b06316, 2017.

---

## Author Comment (AC2) · 9 May 2021

**Response to the anonymous referee #2 (ACP-2020-1143)**

We thank anonymous referee #2 for his/her positive review of this work and relevant comments. Anonymous referee #2's comments/suggestions are given below in bold followed by our replies.

**1) Line 80: Could add reference to e.g. Michoud et al., (2014) which is a more recent specific study on HONO and $NO_y$ relevant to an urban France location (Paris)**

Thanks for the reference, it is now added in the revised manuscript along the Huang et al., (2017) reference.

**2) From what I understand, Eq.(3) assumes that NO $\rightarrow$ NO$_2$ conversion is dominated by O$_3$? Do the authors have information available to clarify this here? i.e. is the observed NO/NO$_2$ ratio close to that given by the Leighton ratio approximation based on NO$_2$ photolysis and NO+O$_3$? Is the discussion on lines 370-373 also relevant here?**

We agree that the overall presentation of this part was not clear. Eq.(3) is developed in Li et al., (2020) assuming that NO conversion to $NO_2$ is dominated by ozone. The main problem was that our detailed discussion of $NO_x$ chemistry and isotopic transfers in the 'Oxygen isotopic composition' section was placed after the analysis of nitrogen isotopic results which actually requires a discussion of $NO_x$ chemistry in order to be easily understood. In addition, the NO + $RO_2$ pathway was neglected in the analysis of nitrogen isotopic results whereas it was taken into account in the analysis of oxygen isotopic results. It is possible to account for the NO + $RO_2$ pathway in the estimation of the nitrogen isotopic fractionation during the day using a new equation linking the nitrogen isotopic fractionation to the oxygen isotopic anomaly. Since this new expression for daytime nitrogen isotopic fractionation contains $\Delta^{17}O$ variables, the discussion of $\Delta^{17}O$ (oxygen isotopic composition) must be presented before the discussion of $\delta^{15}N$ (nitrogen isotopic composition). Furthermore, we also add for completion a more general expression for the nighttime nitrogen isotopic fractionation. In summary, in order to make things clearer and more consistent, we reorganise parts of the paper and accounted for the NO + $RO_2$ pathway in the daytime nitrogen isotopic fractionation analysis. First, the 'Oxygen isotopic composition' section (in particular, the discussion of $NO_x$ chemistry and $\Delta^{17}O$) is now before the 'Nitrogen isotopic composition' section. Second, extended expressions for estimating the daytime and nighttime nitrogen isotopic fractionation are now provided. Derivations of these expressions following Li et al., 2020 are now provided in the new appendix C. The interesting point of the day-ime expression is that, despite that the absence of $RO_2$ measurements, the NO + $RO_2$ pathway can be accounted for in the estimation of nitrogen isotopic fractionation using $\Delta^{17}O$ measurements.

**3) Line 267: Is there any kinetic / mechanistic theoretical explanation that supports the suggestion that NO+RO₂ fractionation is similar to that of NO+O₃? Might be worth mentioning here if so.**

Bigeleisen and Wolfsberg (1957) have laid the foundations of the kinetic isotope fractionation theory within the framework of the transition state theory. Briefly, as a molecule vibrational frequency depends (inversely) on the mass of its atoms, zero point energy (ZPE) of isotopologues differs: the molecule with the heavier isotope has a lower ZPE than the molecule with the lighter isotope. As a result, the dissociation energy is lower for the light isotopologues which facilitates the reaction and increases its reaction rate compared with the heavier isotopologue resulting into an enrichment of the product in the lighter isotope compared to the residual reactant. However, "inverse kinetic isotope fractionation" can also occur leading to an enrichment of the product in the heavier isotope. The range of these isotopic fractionations is defined by the ratio of rate constants for the isotopologue specific reactions.

Previous studies found that the $NO + O_3$ reaction falls within the family of "normal kinetic isotope fractionation" with the $NO_2$ produced being depleted in $^{15}N$ (Walters and Michalski, 2016) compared to residual reactant NO. To our knowledge, no such experiment has been carried out for the $NO + RO_2$ reaction. Nonetheless, considering the very close, and both very low, activation energies for the reaction $NO + O_3$ and $NO + RO_2$ (2.60 kcal mol$^{-1}$ and $-0.71$ kcal mol$^{-1}$, respectively), it is quite likely that the fractionation factors of these two reactions are similar. But we agree it is not proven. In the near future, we plan to carry out experiments in an atmospheric simulation chamber in order to access precisely the nitrogen fractionation factor associated with the reaction $NO + RO_2$. This work will improve our comprehension of nitrogen isotopes fractionations during the atmospheric nitrogen cycle and thus to better quantify $NO_x$ sources from $\delta^{15}N$ measurements.

**Also, can the authors comment on the similarity of the chemical environment in the Li et al., (2020) study to their study location?**

Li et al., (2020) experimental conditions are comparable to multiple tropospheric environments (from clean to polluted sites). Indeed, $NO_x$ and $O_3$ concentrations generated into the simulation chamber under UV-light ($NO_2$ photolysis rate equivalent to dawn) range between 0 to a few dozen of nmol mol$^{-1}$. This suggests that the fractionation factors determined by Li et al. (2020) can be used for our environment. Additionally, while Li et al., (2020) have not determined the temperature dependency of fractionation factors, the daytime temperature variability during our sampling period seems to be too small (from 10 to 17 °C) to have a significant impact on nitrogen fractionations and close enough to Li et al. (2020) experimental conditions (room temperature).

**4) Line 292-299: Comparison with COVID lockdown satellite study seems a bit tenuous (i.e. comparing NO₂ column change over a large city to limited time measurements here). Are there other estimates of traffic contributions to urban NOₓ that could be compared?**

As also noted by referee #1, the comparison to satellite data was a step too far and has been removed now. We now report a 2016 $NO_x$ emissions inventory of the Grenoble urban area. According to this survey, 52 % of $NO_x$ emission are attributed to traffic. This estimation is in good agreement with the value estimated by the new isotopic mixing model (57 ± 8) % (model simulation carried out to estimate the relative contribution of $NO_x$ sources from our nitrogen isotopic measurements). The relatively small difference of 5 % can be attributed to differences in the weather, the location of the sampling site, season and so on. See our replies to comments 3) and 4) from referee #1 for more details about our new estimation of the $NO_x$ sources using the Bayesian isotope mixing model SIAR with updated $\delta^{15}N(NO_x)$ source emission values.

**5) Line 325: The reaction $NO_2 + OH$ to produce $HNO_3$ is termolecular, involving a third body. i.e. $NO + OH + M \rightarrow HNO_3 + M$. See e.g. Atkinson et al., (2004).**

Thanks for pointing out this error, this has been corrected.

**6) Line 369: Derivation of $[RO_2]$ and discussion. The diurnal behaviour of the derived $RO_2$ seems surprising. What is assumed for $k_{NO+RO2}$ in Equation 10? How sensitive is $RO_2$ to the assumed speciation of $RO_2$? i.e. is the value assumed to simply be that for $HO_2$ or is there some weighting for an assumed VOC mixture, and does this matter much? In general it might be useful to provide a Table (in the Appendix?) of values and sources of rate constant values used.**

We agree. The dispersion partly originates from measurements uncertainties, which may not be accurate enough for our level of analysis. According to the literature, $RO_2 + NO$ reactions are relatively fast and do not vary significantly with the nature of the alkyl group (e.g. $k_{NO+CH3O2, 289} = 7.5 \times 10^{-12}$ cm$^3$ molecule$^{-1}$ s$^{-1}$ and $k_{298} = (8-9) \times 10^{-12}$ cm$^3$ molecule$^{-1}$ s$^{-1}$ for other alkyl groups; Atkinson et al., 2006, 2004). In comparison, $HO_2 + NO$ reaction has a slightly lower rate constant of $k_{HO2+NO, 298} = 2.8 \times 10^{-12}$ cm$^3$ molecule$^{-1}$ s$^{-1}$ but it does not impact $RO_2$ concentration calculated with Eq.(6) to more than ± 2 pptv. Consequently, we consider $RO_2$ and $HO_2$ as one group ($RO_2$) and calculated the corresponding rate constant values according to the rate constant expression reported for the $NO + CH_3O_2$ reaction in Atkinson et al., (2006): $k_{NO+RO2} = 2.3 \times 10^{-12} \exp (360/T)$ $^{12}$ cm$^3$ molecule$^{-1}$ s$^{-1}$. As you recommended, we have added an Appendix D reporting a table with the rate constant expressions used for the calculation in our study.

**Typographical errors**
**7) Line 57: ang = and**

Thank you for pointing this typing mistake out.

In the marks reviewed version, you will find in red the main modifications from the first version following your comments and the ones of reviewer #1. We also added two appendixes (C and D). The first appendix presents the derivations of the more general nitrogen isotopic fractionation equations and the second appendix provides a table of the kinetic constants we use for our calculations.

**Reference**

Atkinson, R., Baulch, D. L., Cox, R. A., Crowley, J. N., Hampson, R. F., Hynes, R. G., Jenkin, M. E., Rossi, M. J., and Troe, J.: Evaluated kinetic and photochemical data for atmospheric chemistry: Volume I - gas phase reactions of $O_x$, $HO_x$, $NO_x$ and $SO_x$ species, 4, 1461–1738, https://doi.org/10.5194/acp-4-1461-2004, 2004.

Atkinson, R., Baulch, D. L., Cox, R. A., Crowley, J. N., Hampson, R. F., Hynes, R. G., Jenkin, M. E., Rossi, M. J., and Troe, J.: Evaluated kinetic and photochemical data for atmospheric chemistry: Volume II? gas phase reactions of organic species, 432, 2006.

Huang, R.-J., Yang, L., Cao, J., Wang, Q., Tie, X., Ho, K.-F., Shen, Z., Zhang, R., Li, G., Zhu, C., Zhang, N., Dai, W., Zhou, J., Liu, S., Chen, Y., Chen, J., and O'Dowd, C. D.: Concentration and sources of atmospheric nitrous acid (HONO) at an urban site in Western China, Science of The Total Environment, 593–594, 165–172, https://doi.org/10.1016/j.scitotenv.2017.02.166, 2017.

Li, J., Zhang, X., Orlando, J., Tyndall, G., and Michalski, G.: Quantifying the nitrogen isotope effects during photochemical equilibrium between NO and $NO_2$: implications for $\delta^{15}N$ in tropospheric reactive nitrogen, 20, 9805–9819, https://doi.org/10.5194/acp-20-9805-2020, 2020.

---

## Author Response (AR2)

**Response to the editor comments (ACP-2020-1143)**

**Dear Jan Kaiser,**

We thank you for your acceptance for publication and your relevant final comments. You fill find below your comments in bold followed by our answers.

**1) Fig. 1 axis labels: Please add "amount of ..." before x- and y-axis labels.**

Revised as requested.

**2) l. 129: $\Delta$ (17O) uncertainty is most likely an overestimate because the uncertainties in $\delta$ (17O) and $\delta$ (18O) are correlated. This is confirmed by the Appendix B1. Please provide a more realistic estimate of the $\Delta$ (17O) uncertainty.**

As we don't have reference nitrite standards with a known 17O anomaly, we estimate the  $\Delta$ (17O) uncertainty using uncertainty on  $\delta$ (18O) and  $\delta$ (18O). Overall uncertainty on  $\delta$ 17O and  $\delta$ 18O are estimated to be ± 1.1 and ± 2.5 ‰, respectively, resulting from measurement uncertainty and NO2- storage uncertainty. Using the linear approximation of  $\Delta$ (17O) ( $\Delta$ (17O) =  $\delta$ (17O) – 0.52 ×  $\delta$ (18O)), it results a  $\Delta$ (17O) uncertainty of ± 1.7 ‰.

**3) You should remove the trademark (TM) and registered (®) signs (e.g. used after Finnigan, Corning, 2B Technologies). They are marketing symbols and unnecessary in academic writing.**

Revised as requested.

**4) Times of day should be reported using a 24 hour-clock**

(https://www.atmospheric-chemistry-and-physics.net/submission.html#math)

Revised as requested in Table 1 and Table 3.

**5) l. 190: "keeps", "reaches", "maximum" (singular), not maxima (plural)**

Thank you for pointing this mistake out.

**6) l. 261: 30 pmol mol-1 would be a very low NOx concentration - please check this statement.**

This value is stated in Seinfeld and Pandis, 2006, page 229. Additionally, below a figure taken from Monks, 2005 which shows the balance point between net ozone production and destruction. The compensation point between A and B is between 30 and 100 pptv. We modified the value stated in the manuscript with "above 30-100 pmol mol-1".

**Figure:** Schematic representation of the dependence of the net ozone  $(N(O_3))$  production (or destruction) on the concentration of NOx. The magnitudes reflect clean free tropospheric conditions (from Monks, 2005).

**7) l. 299: The last term should be (6.6±1.5) ‰.**

Thank you for pointing this mistake out.

8) l. 368 & l.681 : Please add unit K after 2470, 1310 and 360 and remove "molecule-1" ("molecule" is not a unit). Revised as requested.

9) At various places (e.g. l. 167 to 187) you use the term "concentration", but refer to amount fractions. In Table 1 and Fig. 2, l. 159 and 162 you call these amount fractions "mixing ratios". The term "mixing ratio" is ambiguous because it can refer to amount fractions, mass fractions and so-called reduced concentrations (at a specific temperature and pressure, eg. 273.15 K and 1bar).

Instead of "amount fraction", the deprecated term "mole fraction" is often used although this would be equivalent to referring to a mass fraction as "kilogram fraction".

You should preferably use the term "amount fraction", but provided you define the term appropriately, I can accept "mixing ratio" or "mole fraction" as well.

Thank you for pointing this out. The paper has been revised as requested using the term mixing ratio and defined as mole fraction.

---

## Author Response (AR3)

**Response to the editor comments (ACP-2020-1143)**

Dear Jan Kaiser,

You fill find below your comments in bold followed by our answer.

**The $\Delta(^{17}O)$ uncertainty cannot be be calculated from simple error propagation of the uncertainties in $\delta(^{17}O)$ and $\delta(^{18}O)$ because the latter are correlated. The combined error has to take the covariation between $\delta(^{17}O)$ and $\delta(^{18}O)$ into account. This is confirmed by the Appendix B1 (l. 555 and Fig. B1) where you state that the uncertainties for nitrite standards are 0.8 ‰ $\delta(^{17}O)$ and 1.8 ‰ $\delta(^{18}O)$. A simple error propagation would give a $\Delta(^{17}O)$ uncertainty of 1.2 ‰, but the actual uncertainty is only 1/4 of that (0.3 ‰). 0.3 ‰ would therefore be a more realistic estimate of the overall uncertainty.**

Thanks for noticing this error. We revised the $\Delta(^{17}O)$ uncertainty considering the dependency of $\delta(^{17}O)$ and $\delta(^{18}O)$ as recommended and following the error propagation expression:

$$u(\Delta^{17}O) = \sqrt{u(\delta^{17}O)^2 + 0.52^2 \times u(\delta^{18}O)^2 - 2 \times 0.52 \times cov(\delta^{17}O, \delta^{18}O)}$$

with $cov(\delta^{17}O, \delta^{18}O) = u(\delta^{17}O) \times u(\delta^{18}O) \times corr(\delta^{17}O, \delta^{18}O)$

Considering $u(\delta^{17}O) = 0.8$ ‰, $u(\delta^{18}O) = 1.8$ ‰ and $corr(\delta^{17}O, \delta^{18}O) = 0.9645$ (see figure below), we obtain a $\Delta(^{17}O)$ uncertainty of 0.3 ‰.

[Figure]

**Figure:** Correlation plot of $\delta^{17}O$ and $\delta^{18}O$ differences between our measurements of RSIL standards (prepared in the KOH/guaiacol eluted matrix) and their certified reference values over three weeks of storage.

**Also, you stated on l. 555 that "If the deviation is constant, it means that the isotopic signal is not degraded with time and its standard deviation is considered as the uncertainty in our $\delta^{17}O(NO_2)$ and $\delta^{18}O(NO_2)$ measurements". This means that the latter uncertainties should be 0.8 ‰ for $\delta(^{17}O)$ and 1.8 ‰ for $\delta(^{18}O)$. It is unclear how you arrive at the higher values of 1.1 ‰ and 2.5 ‰, respectively, further down (l. 561).**

We obtained higher values because we first considered the total $\delta^{17}O$ and $\delta^{18}O$ uncertainties i.e. we added the measurement uncertainty and the storage uncertainty. Nonetheless, we admit it leads to an overestimation of the uncertainty as the measurement uncertainty is already considered into the storage uncertainty.

**Please revise the uncertainty estimates in the manuscript to more realistic, justifiable values.**

The manuscript $\Delta(^{17}O)$ uncertainty has been revised to 0.3 ‰.